# $A^4$-MLRM: Fourfold Attention for Adaptive Hallucination Suppression in Multimodal Large Reasoning Model

## Abstract

Large multimodal reasoning models have recently shown strong ability to solve complex problems by gathering evidence and performing multi-step inference. However, the long reasoning chain makes them more prone to hallucination, that is, generating content that is not supported by the input image or the question. In examining how hallucination arises, we further identify *reasoning drift*: during evidence gathering the model over focuses on entities unrelated to the question, diluting attention on task relevant cues. As a result, previous attention-based methods developed for non-reasoning models often fail to localize the true evidence in reasoning settings. Based on these insights, in this paper, we introduce *AttnRecall*, a metric for assessing visual perception, and present $A^4$-MLRM, a training free, parameter free, and architecture agnostic plugin to hallucination suppression. $A^4$-MLRM uses the model output as a conduit from question to visual tokens for identifying question relevant patches and steer focus to task relevant regions. Remarkably, **without any additional training**, $A^4$-MLRM improves all **reasoning** architectures (including `R1-OneVision`, `Ocean-R1`, `MM-Eureka`, *etc.*) by $1.21\times$ on reasoning benchmarks. When transferred to **non-reasoning** settings, it yields a $1.16\times$ gain. Anonymous codes are available at this link.

## 1 Introduction

In recent years, large multimodal language models (MLLMs) have undergone a paradigm shift from simple image description to unified cross-modal reasoning, giving rise to Multimodal Large Reasoning Models (MLRMs) (Liu et al., 2023c; Huang et al., 2025). These models establish multi-step logical inference chains across visual, textual, and auditory inputs, enabling systematic problem-solving and decision-making in complex, real-world scenarios (Zhao et al., 2024; Yao et al., 2024). To endow them with "thinking" capabilities, practitioners apply supervised fine-tuning or reinforcement learning to a pretrained multimodal backbone, thereby strengthening their inferential strategies and generalization on demanding tasks (Xu et al., 2025; Yu et al., 2024; Guo et al., 2025).

Despite these advantages, stronger reasoning often comes with exacerbated *hallucination*—the generation of content incongruent with the input or factually incorrect (Lu et al., 2025; Zhou et al., 2024; Bai et al., 2024; Liu et al., 2025a). While MLRMs inherit hallucination tendencies from MLLMs, the issue can be amplified in the reasoning setting: the generation pipeline remains language-dominant and produces lengthy deliberations, encouraging over-reliance on linguistic priors and under-utilization of visual evidence (Jiang et al., 2025; Dong et al., 2025a). This modal imbalance weakens visual grounding and increases the likelihood of fabricating non-existent objects or causal explanations not supported by the image (Wu et al., 2025; Fan et al., 2025b).

A related challenge is what we term *reasoning drift* (i.e., attentional diffusion toward task-irrelevant details). Because MLRMs typically marshal numerous "clues" en route to an answer (Yi & Shang, 2025; Wang et al., 2025e), many of which are extraneous to the question, attention can scatter away from visually decisive regions. For example, as illustrated in Fig. 1 (a) and (b), when asked "*Is the batter wearing a helmet?*," the model first enumerate attire attributes (e.g., white pants, black socks), which exceeds the scope of the question and dilutes focus on the helmet. This dispersion undermines attention-based hallucination suppression methods designed for general MLLMs (Park et al., 2025;

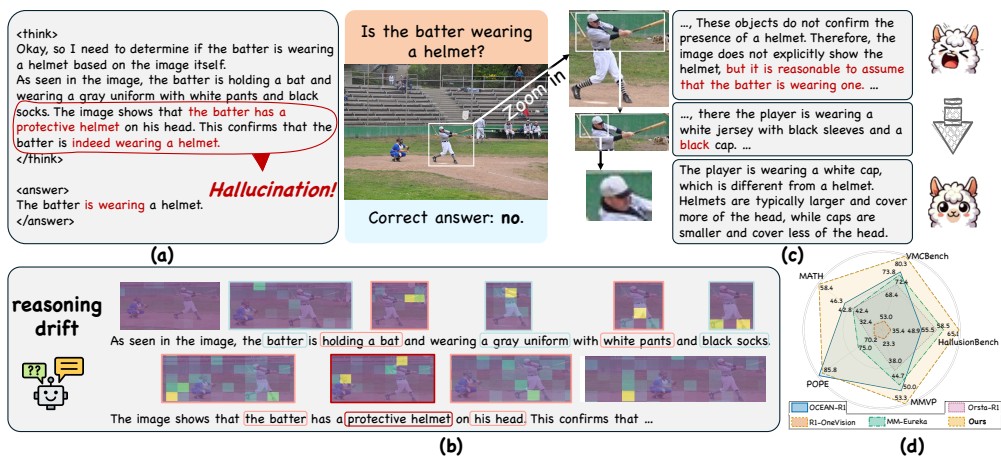

Figure 1: **(a)** Example outputs from a reasoning model; hallucinated content is highlighted in red. **(b)** Attention drift in a multimodal reasoning model, where attention during inference is allocated to task-irrelevant regions. **(c)** An intuitive illustration: by progressively zooming in on task-critical regions, hallucinations (in red) diminish and the answers become correct. **(d)** Radar-chart comparisons among different reasoning models; $A^4$-MLRM is training-free and architecture-agnostic.

An et al., 2025), preventing accurate localization of task-relevant regions and, as a consequence, leaving MLRMs unable to outperform either general MLLMs (Wang et al., 2025b; Liu et al., 2025a) or lean, task-specialized models (Nourbakhsh et al., 2025) on simple tasks.

Existing efforts to mitigate hallucinations in multimodal reasoning models have primarily focused on the *training* stage. One line of work further fine-tunes models on datasets that contain hallucination phenomena or complex reasoning traces (Liu et al., 2025a; Huang et al., 2025; Liu et al., 2025d; Dong et al., 2025a), exposing the model to a broader spectrum of reasoning scenarios so as to calibrate unreliable outputs; another line designs more elaborate training paradigms that explicitly penalize hallucinatory behavior in the loss (Ma et al., 2025b; Duan et al., 2025a; Fan et al., 2025a; Yao et al., 2025; Wang et al., 2025a). However, these approaches require substantial computational and data resources to retrain the model. This invites a natural question: after investing considerable compute and data to endow models with reasoning ability, can we instead intervene *at inference time* and directly leverage the model's own capabilities to alleviate hallucination? This insight seeks a *training-free* solution. Preliminary observations are encouraging: an intuitive example in Fig. 1 (c) shows that progressively constraining the image region under attention—focusing on areas closely tied to the question—markedly reduces hallucination and improves answer accuracy. Building on this intuition, we pose a key scientific question: can we **utilize the multimodal reasoning model's own reasoning and perceptual capacities** to guide attention toward the most question-relevant visual regions during inference, thereby suppressing hallucination?

To approach this question, we note that several studies have begun analyzing attention mechanisms within multimodal reasoning architectures—for example, quantifying cross-modal attention allocation (Liu et al., 2025a; Park et al., 2025) and its evolution over reasoning steps—to probe the drivers of hallucination (Jiang et al., 2025). However, these analyses are largely coarse-grained at the modality level and stop short of finer content- and process-level investigation. Fortunately, multimodal reasoning models expose explicit linguistic chains of thought that convert otherwise latent cognition into an *interpretable reasoning trajectory* (Wang et al., 2025e; Cheng et al., 2025). This observability presents an opportunity: by intervening in attention on the basis of the model's own generated reasoning steps, we can reallocate focus and proactively reduce hallucinations at inference time.

In this paper, we introduce $A^4$-MLRM, an inference-time input–augmentation approach for hallucination suppression. Leveraging the decoder's fourfold attention, $A^4$-MLRM progressively narrows focus from the model's internal layers to the textual query and then to the generated tokens, ultimately isolating the visual evidence most pertinent to the question. Concretely, it addresses: (**A1**) which layer best reflects the model's current understanding; (**A2**) which question tokens are critical to the task; (**A3**) which output tokens answer those key parts; and (**A4**) which visual tokens/regions are

essential to the question. $A^4$-MLRM is training-free and parameter-free, readily transferable across multimodal architectures; its selected evidence can also be reused to adapt non-reasoning models.

Our contributions are as follows:

- **(A1) Average Recall by Attention (ATTNRECALL).** We introduce ATTNRECALL as a model-internal metric to probe visual perception in MLLMs. Using this metric, we observe that in 7B (28-layer) architectures the perception signal peaks at layers **18–24**, reaching $\sim$**50%** ATTNRECALL.
- **(A2–A4) Hallucination suppression in reasoning architectures.** We identify the *reasoning drift* phenomenon in MLRMs and propose $A^4$-MLRM, a training-free mechanism that routes native attention from **question** $\rightarrow$ **output** $\rightarrow$ **visual** to accurately localize task-relevant evidence. On reasoning-oriented hallucination benchmarks, $A^4$-MLRM yields an average $\mathbf{1.25\times}$ improvement on HALLUSIONBENCH and $\mathbf{1.17\times}$ on VMCBENCH, while markedly reducing reasoning drift and strengthening perceptual focus.
- **Transferability to non-reasoning MLLMs.** $A^4$-MLRM is *architecture-agnostic* and transfers to non-reasoning settings, lifting some models (e.g., LLAVA-1.6, R1-ONEVISION) from near chance to the GPT-4V range. On non-reasoning benchmarks, $A^4$-MLRM achieves an **average** accuracy improvement of **+9.3** percentage points.

## 2 PRELIMINARIES

We first define the main notations used in this paper, as summarized in Table 1.

Table 1: **Notations used throughout $A^4$-MLRM.**

| Notation | Description |
|---|---|
| $\mathcal{R}$, $L$ | Multimodal reasoning model (MLRM) and its number of Transformer layers |
| $I$, $q$ | Input image and textual que |
| cat, bbox | Category of an entity in $I$ and its bounding box in the image plane |
| $\mathbf{X}_c$, $\mathbf{X}_s$, $\mathbf{X}_q$, $\mathbf{X}_v$ | Complete input, system prompt tokens, query tokens, and visual tokens |
| $N_c$, $N_s$, $N_q$, $N_v$ | Lengths of $\mathbf{X}_c$, $\mathbf{X}_s$, $\mathbf{X}_q$, and $\mathbf{X}_v$ |
| $\mathbf{X}_{q_{\text{cat}}}$, $\mathbf{X}_{\text{bbox}}$ | Query tokens for category cat; visual tokens within the bounding box bbox |
| $\mathcal{X}_{\text{perc}}$ | Labeled dataset of $(\mathbf{X}_{q_{\text{cat}}}, \mathbf{X}_v, \mathbf{X}_{\text{bbox}}, \text{cat})$ for computing ATTNRECALL |
| $\mathbf{y}_{1:T}$ | Model output token sequence $\{y_1, \ldots, y_T\}$ of length $T$ |
| $\mathbf{A}$ | Attention tensor over inputs, $\mathbf{A} = (A_{t,l,n}) \in [0,1]^{T \times L \times N_c}$ |
| $\mathcal{X}_q^\star$, $\mathbf{y}^\star$, $\mathcal{X}_v^\star$ | Key query, output, and visual tokens selected by A2–A4 |
| zscore($\cdot$), bbox($\cdot$) | $z$-score normalization function; mapping from a region to its bounding box |
| $C_r$, $R_r$ | Cluster $r$ obtained from $\mathcal{X}_v^\star$; crop region corresponding to $C_r$ |

**Multimodal Large Reasoning Models (MLRMs).** We consider decoder-style multimodal systems that connect a pretrained visual encoder to a language model via a projector. The visual encoder and projector map the input image into a sequence of visual tokens $\mathbf{X}_v$. The complete input $\mathbf{X}_c$ is the concatenation of system tokens $\mathbf{X}_s$, visual tokens $\mathbf{X}_v$, and question tokens $\mathbf{X}_q$, i.e., $\mathbf{X}_c = [\mathbf{X}_s, \mathbf{X}_v, \mathbf{X}_q]$, with lengths $N_s$, $N_v$, and $N_q$, and total length $N_c = N_s + N_v + N_q$. At each decoding step $t$, the model samples a token $y_t$ from a conditional distribution $p(y_t \mid \mathbf{X}_c, \mathbf{y}_{<t})$, where $\mathbf{y}_{<t} = \{y_i\}_{i=1}^{t-1}$. Reasoning-oriented MLLMs yield explicit reasoning traces in response to instructions (Liu et al., 2023b; Zhang et al., 2023): in typical settings, the model prints its internal "thinking" between <think> and </think> tags, followed by the final answer.

**Attention Mechanism in MLRMs.** Given an MLRM with a Transformer (Vaswani et al., 2017) decoder of $L$ layers and an output sequence of length $T$, we denote by $\mathbf{A}_{t,l} \in [0,1]^{N_c}$ the attention distribution at layer $l$ and decoding step $t$ over the $N_c$ input tokens in $\mathbf{X}_c$. Concretely,

$$\mathbf{A}_{t,l,:} = \text{softmax}\left(\frac{Q_t^{(l)} K^{(l)\top}}{\sqrt{d_k}}\right), \quad \mathbf{A} \in [0,1]^{T \times L \times N_c}, \quad \sum_{i=1}^{N_c} A_{t,l,i} = 1.$$

We refer to $A$ as the *attention tensor*. This $T \times L \times N_c$ object compactly records, for every output step $(T)$ and layer $(L)$, the normalized distribution over all $N_c$ input-side tokens (system, visual, and question), enabling token-level attribution and layer-wise aggregation.

## 3 MOTIVATIONS

We conduct three targeted study to understand the thinking process of MLRMs and distill three observations (**Obs1–Obs3**) that directly motivate our design in $A^4$-MLRM. Each observation is backed by a minimal, controlled study and quantitative evidence.

**Data context.** We use the POPE dataset built on the *MSCOCO 2014 validation* split (Lin et al., 2014), which provides aligned visual annotations (bounding boxes and categories) and yes/no questions instantiated from the template "Is there a/an {*object*} in the image?" (Li et al., 2023). This pairing lets us regard MSCOCO annotations as ground truth while probing MLRMs' attention behavior at different granularities. Our pilot study uses 1,000 {image, MSCOCO annotations, POPE} triplets; additional explanations are deferred to Appendix D.

**Question to Output Heatmap.** Given an object mention {*object*} detected in the generated output $\mathbf{y}_{1:T}$, we aggregate question to output attention by (i) *distance buckets* around the {*object*} and NOUN_1–NOUN_6 based on token distance—and (ii) POS groups (ADJ/VERB/OTHER). For each question token $x_{q,i}$, we average the question to output attention within each bucket, thereby constructing the question to output heatmap, as shown in Fig. 2 (a).

**Visual to Output Heatmap.** For each ground-truth MSCOCO bounding box (bbox) of a category mentioned in $\mathbf{y}_{1:T}$, we map the box to the model's patch grid and compute mean visual to output attention over the covered *bbox patches*. We then summarize the lateral context by aggregating attention over progressive expansions from the object bbox into *background*. The results are visualized in Fig. 2 (b), with axes centered on the category mention and on the bbox region, respectively.

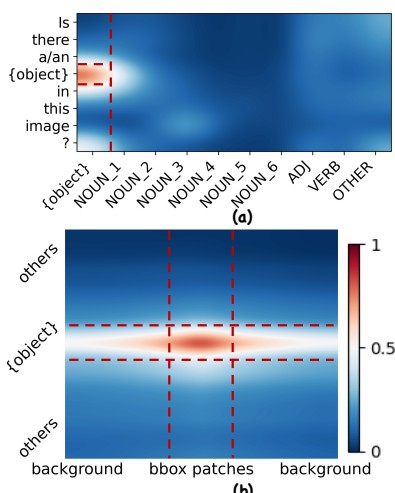

Figure 2: **(a)** Question to Output Heatmap. **(b)** Visual to Output Heatmap. The dashed outline denotes expected attention region, consistent with the heatmap.

> **Obs1:** The MLRM's output primarily addresses the input question token that currently receives the highest attention, as indicated by the **question-to-output attention** weights.

During generation, the model emits clue-like propositions with nominal subjects; attention to the question concentrates only when the current output token is a question-relevant noun. When reasoning drift occurs (attentional diffusion toward task-irrelevant details), question-focused attention diminishes, which we leverage to identify salient output tokens.

> **Obs2:** Key question tokens show larger **attention variance** along the output dimension.

Extending **Obs1**, key question tokens exhibit localized attention peaks along the output axis; consequently their attention traces show larger variance than non-key tokens, providing a efficient, training-free signal for key-token detection.

> **Obs3:** The MLRM's output preferentially describes image regions that currently receives the highest attention, as indicated by the **visual-to-output attention** weights.

Visual–to–output maps reveal a spatial coupling: when the output mentions an entity or attribute, attention concentrates on the corresponding image patches, irrespective of its relevance to the question.

These observations provide a pathway to suppress hallucinations in MLRMs: using the output as a conduit from the input question to the visual tokens. Concretely, we first identify key question tokens, then select question-relevant output tokens, and finally locate the visual tokens most tightly linked to the question. The procedure relies solely on the model's native attention and is applied at inference

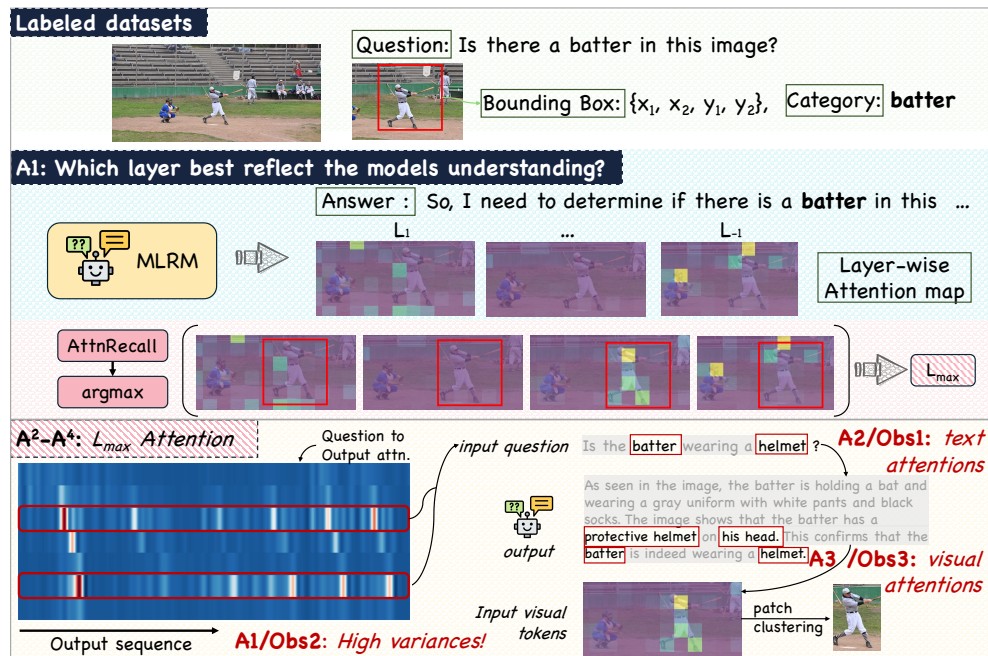

Figure 3: Overview of $A^4$-MLRM. **Top**: labeled data used by A1, the `bbox` and `cat` can either come from COCO annotations or a lightweight model, enabling our pipeline to extend to arbitrary datasets. **Middle**: A1 computes ATTNRECALL layer-wise and selects the layer $L_{\max}$ with the strongest perception. **Bottom**: A2–A4 follow the question → output → visual attention pathway to progressively localize task-relevant visual regions. Best viewed in color.

time, requiring **no additional training or manual annotations**; because it is performed before the final input is fed to the MLRM, it is **architecture-agnostic and transferable**.

## 4 $A^4$-MLRM

In this paper, we present $A^4$-MLRM, an inference-time approach that leverages an MLRM's long reasoning chain to isolate patch tokens critical for answering the question. We introduce the four-fold attention design: A1 identifies the key attention layer in the decoder (Sec. 4.1), and A2–A4 progressively locate pivotal patch tokens via question–output–visual computations (Sec. 4.2). We then describe how the selected visual patch tokens are organized for downstream use (Sec. 4.3). As shown in Fig. 3, the objective of $A^4$-MLRM is to leverage the intrinsic fourfold attention mechanism of MLRM to identify the critical visual tokens required for problem solving, thereby guiding the model to focus more on these elements during reasoning and consequently mitigating hallucinations.

### 4.1 A1: LAYER-WISE PERCEPTION VIA ATTNRECALL.

Perception is a prerequisite for reliable reasoning. The goal of A1 is to assess, for each layer of an MLRM, how well its visual attention captures question-relevant regions. A single image typically contains multiple dense visual clusters, but only a subset is truly informative for answering a given question (e.g., when asked "`Is the batter wearing a helmet?`", the decisive evidence should lie on the batter rather than the catcher). As a precursor to our method, A1 runs an automatic pipeline that measures how much attention each layer allocates to task-critical regions in three steps:

**Query construction.** For each image $I$, we obtain its set of bounding boxes and category labels `cat` from either annotations (e.g., COCO (Lin et al., 2014)) or a segmentation model (e.g., Mask R-CNN (He et al., 2017)). For each category, we instantiate a query using the fixed template $q_{\text{cat}} =$ "`Is there a/an {cat} in this image?`", and tokenize it into question tokens $\mathbf{X}_{q_{\text{cat}}}$.

**Query-aligned regions.** For each $(I, q_{\text{cat}})$ pair, we treat the corresponding bounding box of category cat as the question-relevant region. We encode $I$ into visual tokens $\mathbf{X}_v$ with the visual encoder and projector, and map the bounding box to its covered visual-token indices, denoted by $\mathbf{X}_{\text{bbox}} \subseteq \mathbf{X}_v$.

**ATTNRECALL.** Collecting all such perception-labeled instances yields a set $\mathcal{X}_{\text{perc}} = \{(\mathbf{X}_{q_{\text{cat}}}, \mathbf{X}_v, \mathbf{X}_{\text{bbox}}, \text{cat})\}$. For each item in $\mathcal{X}_{\text{perc}}$, we prepend the system prompt $\mathbf{X}_s$ to form the complete input $\mathbf{X}_c = [\mathbf{X}_s, \mathbf{X}_v, \mathbf{X}_{q_{\text{cat}}}]$, feed it into the reasoning model $\mathcal{R}$, and obtain output tokens $y_1, \ldots, y_T$ and the attention tensor $\mathbf{A} \in [0, 1]^{T \times L \times N_c}$. Let $\mathcal{V}$ denote the visual-token indices within $\mathbf{X}_c$, and let $\mathbf{A}_{t,l,\mathcal{V}} \in [0, 1]^{N_v}$ be the restriction of $\mathbf{A}_{t,l}$ to these visual indices. We evaluate each layer $l$ by how well its visual attention, at the exact output steps where the category token is mentioned, retrieves the ground-truth region $\mathbf{X}_{\text{bbox}}$. For each layer $l \in \{1, \ldots, L\}$, we define

$$\text{ATTNRECALL}(l) = \frac{1}{|\mathcal{X}_{\text{perc}}|} \sum_{\mathbf{X}_c \in \mathcal{X}_{\text{perc}}} \frac{\left| \text{TopK}_{|\mathbf{X}_{\text{bbox}}|} \left( \sum_{t: y_t = \text{cat}} \mathbf{A}_{t,l,\mathcal{V}} \right) \cap \mathbf{X}_{\text{bbox}} \right|}{|\mathbf{X}_{\text{bbox}}|} \tag{1}$$

which directly measures a layer's ability to retrieve object-aligned patches when the object is explicitly mentioned in the model's output. Accordingly, we set $L_{\max} \triangleq \arg\max_{l \in \{1, \ldots, L\}} \text{ATTNRECALL}(l)$ and use $L_{\max}$ as the reference layer for all subsequent attention extraction.

### 4.2 A2-A4: TASK-CRITICAL TOKEN IDENTIFICATION

Building on A1, we have identified the layer with the strongest perception, denoted by $L_{\max}$. In A2–A4, we move to a more general setting: we ask *how to recover task-critical visual regions when neither bounding-box annotations nor fixed question templates are available*. To this end, we leverage the three observations in Sec. 3 and use the attention pathway from question $\rightarrow$ output $\rightarrow$ visual tokens to progressively mine task-relevant visual regions.

**A2: Key query tokens via output-axis variability (Obs. 2).** In *Obs 2*, we find that when the model's output is tightly aligned with the question, the decoder repeatedly allocates attention to a small set of key query tokens. As a result, the attention trajectory of these key tokens along the output axis becomes highly non-uniform, exhibiting larger variance than that of non-key tokens. We therefore collect query-side attention at layer $L_{\max}$ across the entire output axis and retain query tokens whose standardized variability is high. Let $\mathcal{Q} \subseteq \{1, \ldots, N_c\}$ denote the index set of query tokens within $\mathbf{X}_c$. For each query index $n_q \in \mathcal{Q}$ and each decoding step $t \in \{1, \ldots, T\}$, we write

$$A_{t,n_q} = A_{t,L_{\max},n_q}, \qquad \mathbf{a}_{n_q} = (A_{1,n_q}, \ldots, A_{T,n_q}) \in \mathbb{R}^T, \tag{2}$$

where $\mathbf{a}_{n_q}$ is the attention trajectory of the $n_q$-th query token $x_{n_q}$ over the output steps. We compute the variance $\text{Var}(\mathbf{a}_{n_q})$ for each $n_q \in \mathcal{Q}$, and select key query tokens as

$$\mathcal{X}_q^\star = \left\{ x_{n_q} \in \mathbf{X}_q \; : \; \text{zscore}(\text{Var}(\mathbf{a}_{n_q})) \geq \tau_q \right\}, \tag{3}$$

where $\text{zscore}(\cdot)$ denotes the standard score obtained by subtracting the mean and dividing by the standard deviation (Walpole et al., 2011). The indices in $\mathcal{X}_q^\star$ correspond to query tokens whose attention trajectories exhibit high output-axis variability and are thus treated as key query tokens.

**A3: Key output tokens aligned with key query tokens (Obs. 1).** Given the key query token set $\mathcal{X}_q^\star$ from A2, *Obs 1* suggests that when the model's output at step $t$ is closely related to the question, the decoder allocates more attention to these key query tokens. We therefore aggregate, at layer $L_{\max}$, the attention directed from each output step to $\mathcal{X}_q^\star$ and use this signal to select question-relevant output tokens. For each decoding step $t \in \{1, \ldots, T\}$, we define

$$a_t = \frac{1}{|\mathcal{X}_q^\star|} \sum_{x_{n_q} \in \mathcal{X}_q^\star} A_{t,L_{\max},n_q}, \qquad \mathbf{y}^\star = \left\{ y_t \in \mathbf{y} : \text{zscore}(a_t) \geq \tau_o \right\}, \tag{4}$$

where $a_t$ is the average attention mass from output step $t$ to the key query tokens. The set $\mathbf{y}^\star$ thus consists of key output tokens that are most strongly aligned with the key query tokens.

**A4: Key visual tokens mediated by key outputs (Obs. 3).** After identifying the key output tokens $\mathbf{y}^\star$ in A3, we have effectively filtered out output tokens that are irrelevant to the question. ***Obs 3***, together with recent findings (Liu et al., 2025c), suggests that even when the final answer is incorrect, the model often attends to key image regions. We therefore ask: *when emitting these key output tokens, which parts of the image does the model actually focus on?* To answer this, we aggregate, at layer $L_{\max}$, the visual attention associated with $\mathbf{y}^\star$ and use it to select query-relevant visual tokens. Let $\mathbf{X}_v$ denote the visual tokens and $\mathcal{V}$ their index set within $\mathbf{X}_c$. For each index $v \in \mathcal{V}$, we define

$$a_v = \frac{1}{|\mathbf{y}^\star|} \sum_{y_t \in \mathbf{y}^\star} A_{t,L_{\max},v}, \qquad \mathcal{X}_v^\star = \left\{ x_v \in \mathbf{X}_v : \mathrm{zscore}(a_v) \geq \tau_v \right\}, \tag{5}$$

where $a_v$ is the average attention mass that visual token $x_v$ receives over key outputs. The set $\mathcal{X}_v^\star$ thus contains the final task-critical visual tokens that are most strongly associated with the question via the question $\to$ output $\to$ visual attention pathway.

### 4.3 Evidence-Region Construction and Inference-Time Use

**Organization.** Starting from the selected visual tokens $\mathcal{X}_v^\star$, we map each token $x_v^\star \in \mathcal{X}_v^\star$ to its image-plane center $\phi(x_v^\star) \in \mathbb{N}^2$, cluster the point set $\{\phi(x_v^\star)\}_{x_v^\star \in \mathcal{X}_v^\star}$ with DBSCAN (Ester et al., 1996) to obtain clusters $\{C_r\}_{r=1}^R$, and enclose each cluster by an axis-aligned rectangle $R_r = \mathrm{BBox}(C_r)$. The resulting region set $\mathcal{R} = \{R_1, \ldots, R_R\}$ defines crops of the original image. This construction preserves the model's native rectangular interface while consolidating fragmented evidence into a small number of spatially coherent *attention-guided crops*.

**Input modes.** We use these crops in two complementary ways. *Offline:* precompute and cache $\mathcal{R}$ for repeated evaluation or families of similar images. *Online:* a two-stage inference pipeline—**Stage 1** derives $\mathcal{R}$ from the model's attention; **Stage 2** re-invokes the model on $(I; \mathcal{R})$ to refine reasoning with focused evidence. Latency is controlled by limiting the generation length $T$ in Stage 1 and/or the number of retained regions $|\mathcal{R}|$.

## 5 Experiments

In this section, we conduct experiments to address the following research questions:

- **RQ1.** How well does $A^4$-MLRM reduce hallucinations and improve accuracy across diverse reasoning architectures? Which layer best reflects the model's understanding?
- **RQ2.** During generation, Can $A^4$-MLRM mitigate reasoning drift exhibited in Fig. 1, maintaining alignment between the question focus and produced content?
- **RQ3.** How do results change when key settings varied, and specifically, does $A^4$-MLRM's output-mediated patch selection outperform baselines designed for non-reasoning MLLMs?
- **RQ4.** Does $A^4$-MLRM transfer to non-reasoning MLLMs (e.g., LLaVA, Qwen), demonstrating robust, architecture-agnostic generalization?

Table 2: Comparison on reasoning-oriented hallucination benchmarks with and without $A^4$-MLRM. qAcc, fAcc, hAcc, and aAcc denote, respectively, the accuracy *per question pair*, *per figure*, *on hard questions*, and the *overall average*. Case studies are provided in Appendix C.4.

| Model | HallusionBench (Accuracy %) | | | | VMCBench (Accuracy %) | | | | | | Average $\triangle$Acc |
|---|---|---|---|---|---|---|---|---|---|---|---|
| | qAcc | fAcc | hAcc | aAcc | Gen. | Reason. | OCR | Math | Doc&Chart | Overall | |
| R1-OneVision | 8.57 | 12.43 | 31.86 | 35.43 | 56.64 | 45.14 | 65.39 | 32.44 | 52.35 | 52.99 | **+22.20%** |
| +A$^4$ (Ours) | **24.18** | **40.17** | **54.42** | **58.90** | **76.11** | **61.81** | **89.97** | **51.21** | **78.88** | **73.91** | |
| Ocean-R1 | 19.34 | 27.76 | 40.00 | 48.89 | 77.74 | 61.69 | 88.91 | 46.25 | 76.58 | 73.75 | **+10.60%** |
| +A$^4$ (Ours) | **30.11** | **43.35** | **54.88** | **63.51** | **83.51** | **66.99** | **97.19** | **53.36** | **85.11** | **80.32** | |
| MM-Eureka | 23.07 | 31.50 | 49.30 | 58.46 | 76.31 | 58.79 | 90.46 | 42.44 | 76.21 | 72.44 | **+7.26%** |
| +A$^4$ (Ours) | **33.41** | **47.11** | **56.98** | **65.10** | **83.56** | **67.26** | **94.05** | **58.38** | **85.93** | **80.31** | |
| ORSTA-R1 | 21.76 | 28.61 | 45.81 | 55.45 | 71.55 | 55.62 | 85.06 | 42.82 | 72.60 | 68.39 | **+7.30%** |
| +A$^4$ (Ours) | **27.91** | **36.71** | **53.49** | **60.05** | **81.52** | **64.21** | **94.41** | **51.11** | **84.63** | **78.39** | |

## 5.1 EXPERIMENTAL SETUP

We begin with a concise overview of models, datasets, and evaluation metrics; full implementation and protocol details are deferred to the Appendix Section B.2.

**Models.** $A^4$-MLRM is compatible with existing hallucination-mitigation schemes for MLRMs. Accordingly, we apply it to several reasoning models—R1-OneVision-7B (Yang et al., 2025), Ocean-R1-7B (Ming et al., 2025), Orsta-R1-7B (Ma et al., 2025b), and MM-Eureka-7B (Meng et al., 2025)—to assess generality. To test transferability, we also port $A^4$-MLRM to *non-reasoning* MLLMs, including LLaVA-1.6-Mistral-7B (Liu et al., 2024) and Qwen2.5-VL-7B (Bai et al., 2025).

**Datasets & Evaluation Metrics.** We evaluate on four widely used hallucination benchmarks, grouped as follows: *reasoning-oriented*: **1)** VMCBENCH (Zhang et al., 2025c), a unified multiple-choice questions drawn from 20 VQA datasets, e.g., *MathVision* (Wang et al., 2024), *ScienceQA* (Lu et al., 2022), and **2)** HALLUSIONBENCH (Guan et al., 2024) benchmarks image–context reasoning where language hallucinations and visual illusions are entangled; and *perception-oriented*: **3)** POPE (Li et al., 2023) and **4)** MMVP (Tong et al., 2024). We follow each dataset's official evaluation protocol.

Table 3: Per-layer ATTNRECALL (left) and benchmark accuracy on POPE/MMVP with and without $A^4$-MLRM (right). Higher is better.

| Model | ATTNRECALL per Layer (%) | | | | | | POPE (%) | | MMVP (%) | |
| --- | --- | --- | --- | --- | --- | --- | --- | --- | --- | --- |
| | 0 | 6 | 12 | 18 | 24 | 27 | w/o $A^4$ | w/ $A^4$ | w/o $A^4$ | w/ $A^4$ |
| R1-OneVision | 30.78% | 33.68% | 43.11% | **50.60%** | 47.28% | 44.11% | 70.22% | **81.64%** | 23.33% | **46.00%** |
| Ocean-R1 | 31.05% | 32.64% | 48.56% | **53.79%** | 50.92% | 49.23% | **86.77%** | 85.77% | 47.33% | **50.00%** |
| MM-Eureka | 29.30% | 30.57% | 45.45% | 51.06% | **51.38%** | 48.11% | 75.00% | **81.38%** | 44.67% | **50.00%** |
| ORSTA-R1 | 31.14% | 32.38% | 47.55% | 51.66% | **55.32%** | 50.56% | 71.36% | **82.98%** | 38.00% | **53.33%** |

## 5.2 PERFORMANCE WITH $A^4$-MLRM ON REASONING MODELS (RQ1)

To assess hallucination in *reasoning* models, we evaluate each model *before* and *after* applying $A^4$-MLRM on hallucination benchmarks; the aggregated results are reported in Table 2 and Table 3, more results are shown in Appendix C. Based on these tables, we can draw the following findings:

**Finding 1: Consistent gains across models and datasets.** $A^4$-MLRM improves *all* reasoning models across the four hallucination benchmark. Specifically, on the reasoning-oriented hallucination datasets—HALLUSIONBENCH and VMCBENCH—$A^4$-MLRM achieves a **maximum** accuracy gain of **22.20%** and an **average** accuracy gain of **11.84%**.

**Finding 2: Models with different training architectures exhibit similar perceptual capacity.** From Table 3, the ATTNRECALL curves for all models peak at **layer 18 or layer 24** and follow a rise–then–fall pattern. On the perception-oriented hallucination benchmarks—POPE and MMVP—$A^4$-MLRM delivers an **average** absolute gain of **9.31%**.

## 5.3 HALLUCINATION ANALYSIS (RQ2)

To further verify the hallucination–suppression ability of $A^4$-MLRM, we compute two key indicators on the POPE dataset: (1) {*object*} *visual attention*, i.e., the attention mass that falls inside the {*object*} bounding box; and (2) {*object*} *text proportion*, i.e., the fraction of noun tokens in the model output that correspond to {*object*}, we also present examples where the model reduces reasoning drift and strengthens perceptual focus, as shown in Fig. 4. Our key findings are as follows:

**Finding 3:** $A^4$**-MLRM mitigates reasoning drift.** As shown in Fig. 4 (a), after models are equipped with $A^4$-MLRM, the {*object*} *text proportion* in the generated text **increases** markedly across all models. Fig. 4 (b) provide two illustrative cases. *Without* $A^4$-MLRM, the model offers exploratory descriptions of irrelevant regions while **overlooking** the truly discriminative area; *with* $A^4$-MLRM, it **directly focuses** on the region that needs to be judged. In Fig. 4 (b-2), for example, the baseline over-attends to non-key areas (*desk → computer setup → laptop → monitor → keyboard → mouse → headphones → notebook*), thereby **missing** the hand in the bottom-right corner.

**Finding 4:** $A^4$**-MLRM reduces perceptual hallucination.** As shown in Fig. 4, $A^4$-MLRM allocates **higher attention to key regions**, enabling the model to attend to areas that were previously hard to perceive. In Fig. 4 (c-2), for instance, $A^4$-MLRM surfaces a faint truck silhouette that humans also found subtle; prior to applying $A^4$-MLRM, all reasoning models predicted "no."

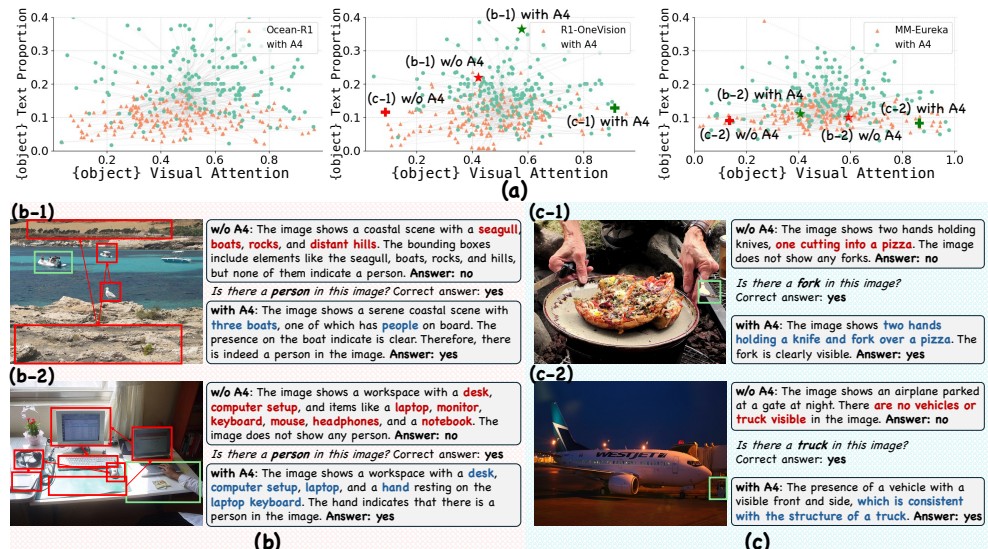

Figure 4: **(a)** Comparative scatter plot of attention allocation versus output mentions for task-relevant objects; **(b)** case study showing that $A^4$-MLRM reduces reasoning drift; **(c)** case study showing that $A^4$-MLRM enhances fine-grained perceptual sensitivity.

## 5.4 COMPONENT STUDIES (RQ3)

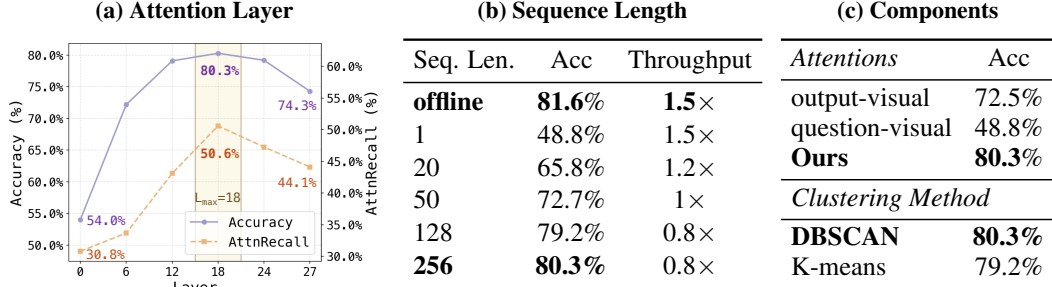

| (a) Attention Layer | (b) Sequence Length | | | (c) Components | |
|---|---|---|---|---|---|
| | Seq. Len. | Acc | Throughput | *Attentions* | Acc |
| | **offline** | **81.6%** | **1.5×** | output-visual | 72.5% |
| | 1 | 48.8% | 1.5× | question-visual | 48.8% |
| | 20 | 65.8% | 1.2× | **Ours** | **80.3%** |
| | 50 | 72.7% | 1× | *Clustering Method* | |
| | 128 | 79.2% | 0.8× | **DBSCAN** | **80.3%** |
| | 256 | **80.3%** | 0.8× | K-means | 79.2% |

Figure 5: Ablation Study. Default settings in the experiments are shown in **bold**.

As discussed in pervious sections, $A^4$-MLRM effectively identifies task-relevant visual evidence. In this section, we examine the *module sensitivity* of $A^4$-MLRM and explain why analogous approaches developed for *non-reasoning* MLLMs do not carry over to *reasoning* models. We perform ablation studies on R1-ONEVISION with POPE; the results are summarized in Figure 5. Our findings are:

**Finding 5: Performance trends align with expectations.** Based on Figure 5 (a–c), the accuracy trend over attention layers mirrors the ATTNRECALL pattern in Table 3. For **sequence length**, in the online setting, a longer Stage 1 output provides richer priors to Stage 2, thus yielding higher accuracy. For **clustering**, DBSCAN is preferable to K-means, as it is less sensitive to noise and can naturally discard isolated noisy tokens when the attention map contains spurious activations; see Appendix C.3 for a detailed discussion.

**Finding 6: $A^4$-MLRM outperforms prior attention-based hallucination suppression methods on reasoning models.** As shown in Figure 5 (c), $A^4$-MLRM improves over attention-only baselines (question→visual or output→visual) developed for non reasoning settings, which are less effective under the long context dynamics of reasoning models.

## 5.5 TRANSFER $A^4$-MLRM TO NON-REASONING MLLMS (RQ4)

To verify the transferability of $A^4$-MLRM, we evaluate MMVP on *non-reasoning* MLLMs. Concretely, we adopt the *offline* setting: attention-guided crops produced by OCEAN-R1 are fed to QWEN2.5-VL and LLAVA-1.6-MISTRAL. Results are summarized in Fig. 6. Our findings are:

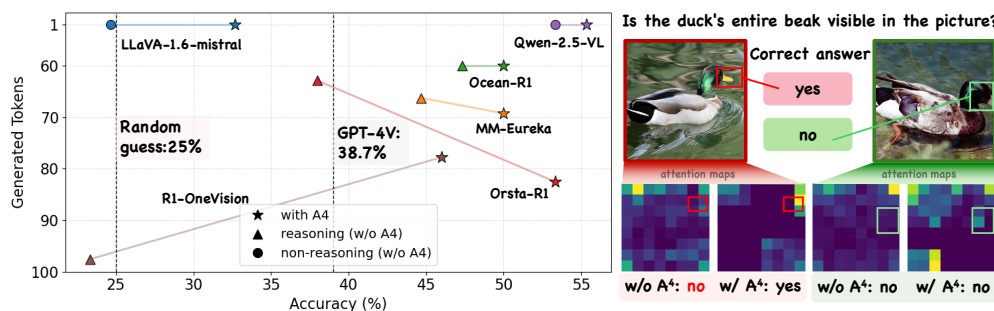

Figure 6: **Left:** Accuracy versus output–token length on MMVP across models; **Right:** attention allocation with and without $A^4$-MLRM.

**Finding 7: $A^4$-MLRM broadly strengthens both reasoning and non-reasoning MLLMs.**
MMVP is designed to probe visual defects in multimodal models. As shown in Fig. 6, $A^4$-MLRM consistently improves performance; in particular, it moves several models—LLAVA-1.6, R1-ONEVISION, and ORSTA-R1—from near random guessing toward the GPT-4V level. In the qualitative examples of Fig. 6, without $A^4$-MLRM the models barely attend to the key region of the question, whereas with $A^4$-MLRM they shift focus to that region.

## 6 RELATED WORK

**Multimodal Large Reasoning Models (MLRMs).** Early works extended chain-of-thought (CoT) reasoning to vision-language models through supervised fine-tuning and reinforcement learning (RL), with methods like Marco-o1 integrating search and reflection strategies (Zhao et al., 2024). Subsequent approaches enhanced stepwise reasoning via self-refinement (Zhang et al., 2024a) and long-chain data scaling (Xu et al., 2025), while VLMs adopted CoT supervision for visual reasoning (Liu et al., 2023c; Thawakar et al., 2025; Yao et al., 2024). Preference-based RL alignment emerged to reduce factual errors using human feedback (Yu et al., 2024), closed-loop optimization (Zhang et al., 2024d), and reasoning trace comparisons (Dong et al., 2025b). The GRPO paradigm introduced by DeepSeek-R1 established rule-based reward optimization as standard practice (Guo et al., 2025; Liu et al., 2025b; Zhang et al., 2024b; Huang et al., 2025; Wang et al., 2025c; Meng et al., 2025). Two dominant approaches exist: *1) Two-stage SFT+RL pipelines* (e.g., R1-OneVision, Reason-RFT) (Yang et al., 2025; Tan et al., 2025; Zhang et al., 2025a), and *2) Direct large-scale RL training* ("R1-Zero") yielding emergent reasoning (Ming et al., 2025; Wang et al., 2025d). Recent innovations include unified frameworks for joint reasoning/perception (Ma et al., 2025b) and RL-enhanced generative reasoning (Duan et al., 2025b).

**Hallucination in MLRMs.** Extended reasoning chains exacerbate visual hallucinations as models prioritize language priors over visual input (Bai et al., 2024; Liu et al., 2025a). New benchmarks like MIRAGE quantify reasoning-specific errors through granular metrics (Dong et al., 2025a). Mitigation strategies feature: *1) Explicit grounding:* Region recognition–reasoning–refinement (VLM-$R^3$) first localizes key image regions before text generation (Jiang et al., 2025); Chain-of-Focus employs RL to adaptively zoom into salient regions (Zhang et al., 2025b); GRIT interleaves bounding-box references with each CoT step (Fan et al., 2025b). *2) Post-hoc verification:* The "Look Twice" approach re-encodes image memory features mid-generation and aligns them with generated text to identify and correct inconsistencies (Zou et al., 2024). *3) Preference optimization:* Entity-centric multimodal preference optimization (EMPO) penalizes descriptive mismatches with visual entities during training (Wu et al., 2025). *4) Adaptive reasoning policies:* The "Think-or-Not" strategy evaluates whether multi-step CoT is necessary, skipping or invoking detailed reasoning to prevent unnecessary hallucination (Wang et al., 2025a).

## 7 CONCLUSION

In this paper, we introduce $A^4$-MLRM, a new hallucination suppression approach for multi-modal reasoning models that is training free, parameter free, and architecture agnostic, thus incurring minimal deployment cost. Concretely, $A^4$-MLRM leverages fourfold attention along the question→output→visual pathway to identify question relevant visual patches and steer focus toward task relevant regions, enabling more precise inference. Across both reasoning and non reasoning settings, $A^4$-MLRM yields an average improvement of **15.1** percentage points.

## REPRODUCIBILITY STATEMENT

We provide an anonymous code repository and the complete set of generated outputs. The experimental setup—datasets, models, and evaluation protocols—is summarized in Section 5.1 and detailed in Appendix B, with links to openly available sources. The exact inference configurations required to reproduce results, including prompt templates and decoding parameters, are specified in Section B.2 and Section 5.4. Our method is inference-only (no fine-tuning), and all third-party datasets and checkpoints are public. These materials collectively enable end-to-end reproduction of our findings.

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

CONTENTS

## A LIMITATIONS AND FUTURE WORK

**Limitations.** While $A^4$-MLRM efficiently mitigates hallucination in MLRMs, several limitations merit acknowledgment and point to concrete remedies. **(i) Two-stage inference.** Owing to current reasoning-model architectures, $A^4$-MLRM operates in two stages (attention mining then focused re-inference). A learned controller that adaptively decides *when* to trigger $A^4$-MLRM and *how much* context to generate (e.g., output length for Stage 1) could reduce overhead and retain quality in a single pass. **(ii) Rectangular crops.** Our evidence regions are axis-aligned; objects, however, are often non-rectangular, so rectangular cropping may include extraneous background.

**Future work.** **(i) Single-pass integration.** Post-train a lightweight policy to fuse $A^4$-MLRM into one forward pass, enabling adaptive, example-dependent activation and scope (triggering and output length) within the model's native decoding; reinforcement learning or adaptive-compute objectives are natural fits. **(ii) Beyond rectangles.** Explore mask- or token-level selection: derive instance/part masks with promptable segmentation to form shape-accurate evidence, or re-inject only selected visual tokens (retaining their original positional encodings) while dropping or down-weighting others, thus avoiding commitment to a particular crop geometry.

## B EXPERIMENT DETAILS

### B.1 BASELINES

**Models.** Because $A^4$-MLRM is training-free and architecture-agnostic, we integrate it at inference into four recent multimodal reasoning models without altering their weights or decoding. (i) **R1-OneVision** (Yang et al., 2025)—a cross-modal formalization pipeline that converts visual content into structured textual representations before language reasoning, trained with SFT and RL. (ii) **Ocean-R1** (Ming et al., 2025)—an open RL-enhanced VLM whose public artifacts indicate a two-stage recipe (reasoning first, then visual perception) with released weights/data. (iii) **Orsta** (Ma et al., 2025b)—a family of VLMs post-trained via the *V-Triune* unified RL system to improve both perception and reasoning. (iv) **MM-Eureka** (Meng et al., 2025)—a rule-based (R1-style) reinforcement learning approach extended to multimodal reasoning with an open pipeline. To assess transferability, we further deploy the $A^4$-MLRM-derived visual cropping prior on two *non-reasoning* MLLMs: (v) **Qwen2.5-VL** (Bai et al., 2025)—a general-purpose vision–language model series (3B/7B/72B) with an efficient ViT and official instruction-tuned checkpoints; and (vi) **LLaVA-1.6-Mistral** (Liu et al., 2024)—an instruction-tuned VLM pairing a vision encoder with Mistral-7B, documented with improved training data and dynamic high-resolution support.

**Datasets.** We evaluate hallucination on four benchmarks. **POPE** (Li et al., 2023) targets object-level hallucination via yes/no "object presence" queries and contains three standard splits (popular/adversarial/random). **VMC-Bench** (Zhang et al., 2025c) unifies 20 VQA datasets into a single multiple-choice suite (9,018 questions) spanning diverse domains. **MMVP** (Tong et al., 2024) focuses on nine basic visual patterns and "CLIP-blind pairs," probing failure modes that induce incorrect answers or overconfident, hallucinatory explanations. **HallusionBench** (Guan et al., 2024) is an image–context reasoning benchmark diagnosing entangled language hallucination and visual illusion, consisting of 346 images and 1,129 expert-crafted questions with control structures for analysis. For **POPE** we report accuracy using the *lmms-eval* (Zhang et al., 2024c) implementation. For **VMC-Bench**, **MMVP**, and **HallusionBench** we follow each benchmark's official evaluation pipeline.

### B.2 EVALUATION SETUP DETAILS

**Prompt templates.** For *yes/no* questions (POPE, MMVP, HallusionBench) and *multiple choice* questions (VMCBench), we use the following templates:

> You FIRST think about the reasoning process as an internal monologue and then provide the final answer. The reasoning process MUST BE enclosed within <think></think> tags. The final answer MUST BE a single word enclosed in <answer></answer> tags. Let's think more.
> {question}

> You FIRST think about the reasoning process as an internal monologue and then provide the final answer. The reasoning process MUST BE enclosed within <think></think> tags. The final answer must be an option letter from the given choices, enclosed in <answer></answer> tags. Let's think more.
> {question}

**Evaluation.** During evaluation, we parse the model output with a regular expression to extract the content enclosed in `<answer>...</answer>` and compare it with the ground truth.

**Generate config.** We use the same decoding settings across all datasets; any parameter not listed remains at its default.

| Parameter | Value |
| --- | --- |
| min_new_tokens | 50 |
| max_new_tokens | 1024 |
| do_sample | True |

## C  MORE EXPERIMENTAL RESULTS

### C.1  MORE RESULTS ON POPE AND VMC-BENCH

**Evaluation Setup.**  We report detailed results across multiple benchmarks following each dataset's official protocol. For POPE, we report accuracy and F1 on the *Random*, *Popular*, and *Adversarial* splits (Li et al., 2023). For VMCBENCH, we report per-subset and overall accuracy (Zhang et al., 2025c). For MMVP, we follow the authors' evaluation on CLIP-blind visual patterns (Tong et al., 2024).

**VMCBENCH Subsets.**  We adopt VMCBENCH's four capability-oriented subsets and their official composition (Zhang et al., 2025c): **(1) General**: *VQAv2* (Goyal et al., 2017), *GQA* (Hudson & Manning, 2019), *VizWiz* (Gurari et al., 2018), *OK-VQA* (Marino et al., 2019), *A-OKVQA* (Schwenk et al., 2022); **(2) Reasoning**: *MMMU* (Yue et al., 2023), *MathVista* (Lu et al., 2023), *MathVision* (Wang et al., 2024), *MMStar* (Chen et al., 2024); **(3) OCR**: *TextVQA* (Singh et al., 2019), *OCR-VQA* (Mishra et al., 2019); **(4) Doc & Chart**: *DocVQA* (Mathew et al., 2021), *TableVQA-Bench* (Kim et al., 2024), *InfographicVQA (InfoVQA)* (Mathew et al., 2022), *AI2D* (Kembhavi et al., 2016), *ChartQA* (Masry et al., 2022).

**Reasoning subset members.** *MMMU*: college-level, multi-discipline problems requiring deliberative reasoning (Yue et al., 2023). *MathVista*: mathematical reasoning in visual contexts (charts, plots, geometry, etc.) (Lu et al., 2023). *MathVision*: math problem solving on real-world photos and synthetic scenes (Wang et al., 2024). *MMStar*: broad, multi-skill evaluation of multimodal reasoning (Chen et al., 2024).

**General subset members.** *VQAv2*: balanced open-ended VQA over everyday photos (Goyal et al., 2017). *GQA*: compositional visual reasoning with scene-graph grounding (Hudson & Manning, 2019). *VizWiz*: real-world VQA from blind photographers; noisy images and conversational questions (Gurari et al., 2018). *OK-VQA/A-OKVQA*: knowledge-based VQA requiring external/world knowledge (Marino et al., 2019; Schwenk et al., 2022).

**OCR subset members.** *TextVQA*: reading text in the wild for VQA (Singh et al., 2019). *OCR-VQA*: VQA by reading textual content in images (book covers, storefronts, etc.) (Mishra et al., 2019).

**Doc & Chart subset members.** *DocVQA*: document understanding for VQA (forms, invoices, pages) (Mathew et al., 2021). *TableVQA-Bench*: multi-domain table QA benchmark for visual tables (Kim et al., 2024). *InfographicVQA*: QA over complex infographics requiring reading and reasoning (Mathew et al., 2022). *AI2D*: diagram understanding and diagram-based QA (Kembhavi et al., 2016). *ChartQA*: QA over charts requiring numerical and logical reasoning (Masry et al., 2022).

Table 4: Accuracy (%) on selected VMCBENCH. Including: MMMU, MathVista, MMStar, AI2D, ScienceQA, SEEDBench, MM-Vet, MathVision, TableVQABench (TVQA), RealWorldQA (RWQA). Rows are paired as baseline vs. +A$^4$ for each model.

| Model | MMMU | MathVista | MMStar | AI2D | ScienceQA | SEEDBench | MM-Vet | MathVision | TVQA | RWQA |
|---|---|---|---|---|---|---|---|---|---|---|
| R1-OneVision | 36.30 | 36.63 | 40.38 | 46.70 | 61.54 | 54.81 | 48.12 | 28.99 | 48.31 | 46.79 |
| +A$^4$ (Ours) | **50.49** | **67.00** | **55.96** | **68.06** | **82.35** | **72.35** | **74.29** | **36.41** | **73.79** | **54.62** |
| Ocean-R1 | 56.28 | 60.89 | 58.81 | 74.94 | 80.09 | 73.58 | 73.85 | 32.35 | 69.02 | 60.09 |
| +A$^4$ (Ours) | 55.42 | **69.50** | **62.53** | **79.68** | **84.39** | **80.99** | **79.44** | **36.97** | **79.09** | **66.67** |
| MM-Eureka | 51.20 | 58.42 | 56.06 | 71.07 | 79.64 | 72.84 | 73.38 | 27.42 | 68.02 | 54.59 |
| +A$^4$ (Ours) | **60.58** | **74.75** | **63.66** | **80.41** | **83.48** | **77.78** | **85.61** | **41.78** | **80.41** | **56.42** |
| ORSTA-R1 | 45.43 | 51.49 | 51.31 | 66.06 | 71.04 | 70.37 | 67.67 | 33.17 | 67.65 | 57.57 |
| +A$^4$ (Ours) | **58.89** | **68.50** | **62.71** | **74.94** | **79.41** | **76.30** | **85.82** | **34.22** | **77.03** | **63.07** |

**Additional sets in VMCBENCH.** *MM-Vet*: comprehensive capability evaluation of MLLMs, frequently used as a generalization stress test (Yu et al., 2023). *SEED-Bench*: multi-domain, fine-grained evaluation for perception and reasoning (Li et al., 2024). *RealWorldQA*: real-world spatial understanding; images largely from vehicles with verifiable Q&A (xAI, 2024).

**Thinking length.** We measure the average output token length on the "general" and "reasoning" subsets of VMC-Bench. With $A^4$-MLRM, the average output length on the reasoning subset increases from 197.22 to 302.88 tokens (a 53.57% increase), while on the general subset it only increases from 134.43 to 136.10 tokens (a 1.24% increase). This further supports that $A^4$-MLRM reduces the perceptual burden and lets the model use more capacity for reasoning.

Table 5: Accuracy (%) on the remaining VMCBENCH datasets: TextVQA, InfoVQA, DocVQA, OCRVQA, VizWiz, ChartQA, GQA, A-OKVQA, OKVQA, VQAv2. Rows are paired as baseline vs. +A$^4$ for each model.

| Model | TextVQA | InfoVQA | DocVQA | OCRVQA | VizWiz | ChartQA | GQA | A-OKVQA | OKVQA | VQAv2 |
|---|---|---|---|---|---|---|---|---|---|---|
| R1-OneVision | 64.72 | 56.36 | 58.56 | 66.06 | 64.18 | 56.65 | 61.86 | 56.71 | 63.95 | 66.44 |
| +A$^4$ (Ours) | **90.56** | **77.14** | **94.44** | **89.38** | **84.42** | **82.11** | **80.93** | **80.24** | **81.98** | **84.03** |
| Ocean-R1 | 91.01 | 71.19 | 92.00 | 87.31 | 86.86 | 75.92 | 80.68 | 81.65 | 86.67 | 83.56 |
| +A$^4$ (Ours) | **97.75** | **82.16** | **100.00** | **96.63** | **90.91** | **84.63** | **88.75** | **89.41** | **92.35** | **88.89** |
| MM-Eureka | 88.54 | 73.73 | 88.42 | 91.71 | 83.09 | 79.36 | 82.40 | 79.06 | 85.68 | 83.56 |
| +A$^4$ (Ours) | **94.83** | **82.75** | **96.88** | **93.26** | **90.44** | **88.99** | **87.04** | **86.82** | **93.33** | **87.96** |
| ORSTA-R1 | 83.37 | 70.43 | 82.20 | 86.79 | 82.99 | 76.15 | 72.86 | 72.94 | 75.80 | 81.02 |
| +A$^4$ (Ours) | **95.28** | **86.71** | **97.97** | **93.52** | **89.18** | **86.70** | **81.66** | **80.71** | **88.89** | **87.27** |

Table 6: POPE: accuracy (%) and F1 by split (Adversarial / Popular / Random / Average).

| Model | Adversarial | | Popular | | Random | | Average | |
|---|---|---|---|---|---|---|---|---|
| | Acc | F1 | Acc | F1 | Acc | F1 | Acc | F1 |
| R1-OneVision | 69.20% | 0.6629 | 70.33% | 0.6711 | 71.12% | 0.6769 | 70.22% | 0.6703 |
| +A$^4$ (Ours) | **80.90%** | **0.7731** | **81.60%** | **0.7825** | **82.43%** | **0.7881** | **81.64%** | **0.7805** |
| Ocean-R1 | **85.53%** | **0.8481** | **86.80%** | **0.8592** | **87.97%** | **0.8707** | **86.77%** | **0.8593** |
| +A$^4$ (Ours) | 84.43% | 0.8271 | 86.13% | 0.8458 | 86.70% | 0.8496 | 85.76% | 0.8408 |
| MM-Eureka | 74.17% | 0.7107 | 74.21% | 0.7108 | 76.63% | 0.7308 | 75.00% | 0.7174 |
| +A$^4$ (Ours) | **80.73%** | **0.7726** | **81.17%** | **0.7761** | **82.25%** | **0.7860** | **81.38%** | **0.7782** |
| ORSTA-R1 | 68.31% | 0.6445 | 72.00% | 0.6754 | 73.77% | 0.6866 | 71.36% | 0.6688 |
| +A$^4$ (Ours) | **81.86%** | **0.7895** | **83.39%** | **0.8061** | **83.67%** | **0.8071** | **82.98%** | **0.8009** |

## C.2 RESULTS ON RWQA AND 3DSRBENCH

We evaluate $A^4$-MLRM on RWQA (xAI, 2024) and 3DSRBench (Ma et al., 2025a) using three base models. As shown in Table 7, $A^4$-MLRM consistently improves accuracy on both datasets for all backbones.

Table 7: Results on RWQA and 3DSRBench with and without $A^4$-MLRM.

| Dataset | Setting | Ocean-R1 | ORSTA | MM-Eureka |
|---|---|---|---|---|
| RWQA | w/o $A^4$-MLRM | 63.27% | 65.23% | 60.39% |
| RWQA | with $A^4$-MLRM | 70.33% | 67.84% | 65.49% |
| 3DSRBench | w/o $A^4$-MLRM | 37.66% | 43.44% | 56.25% |
| 3DSRBench | with $A^4$-MLRM | 40.06% | 45.14% | 59.24% |

## C.3 RESULT DISCUSSIONS

In this section, we discuss the ablation results in detail.

**Clustering methods.** We compare K-MEANS and DBSCAN at the patch clustering step. As shown in Table 5, K-MEANS is a partition-based method, forcing every patch to be assigned to some cluster; this makes it sensitive to noise. In contrast, DBSCAN is density-based and permits points outside high-density regions to remain as noise. Practically, K-MEANS tends to absorb noisy patches into clusters, yielding overly large crops, whereas DBSCAN better isolates compact evidence regions. An example is provided in Fig. 7: K-MEANS nearly crops the entire image, while DBSCAN focuses on the key area.

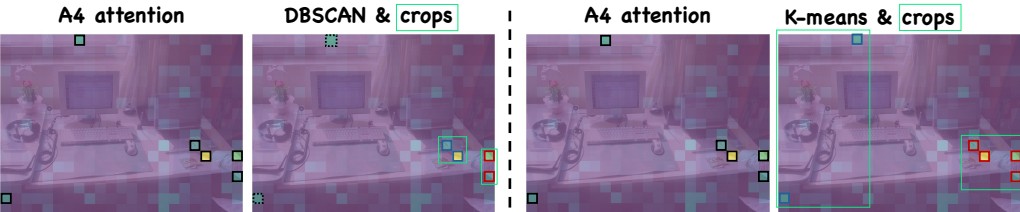

Figure 7: **Case study on clustering methods.** Each subfigure has two parts. *Left:* $A^4$ attention, i.e., a visualization of the attention map, where the $A^4$-selected visual patches $\mathcal{V}_k^\star$ are outlined with black boxes. *Right:* clustering results from different methods. Because K-means is sensitive to noise, the resulting crops include large amounts of question-irrelevant content induced by only two noisy points, whereas DBSCAN avoids this issue.

**Attention choices.** We compare $A^4$-MLRM with two baselines: *output→visual* attention (common in generative pipelines) and *question→visual* attention (input–input cross-attention, common in discriminative settings); see Fig. 8. Due to *reasoning drift*, output→visual alone can be distracted by complex scenes and fail to attend to the question-relevant region. Conversely, question→visual underutilizes the model's training signal, which is primarily aligned to fitting the output sequence, leading to a mismatch and degraded localization.

## C.4 CASE STUDY

We present side-by-side comparisons across multiple reasoning models *with* and *without* $A^4$-MLRM. As shown in Fig. 9 and Fig. 10, red boxes denote the visual crops. $A^4$-MLRM consistently localizes task-relevant regions and produces more accurate answers than the baseline. We also report failure cases in Fig. 11 and Fig. 12, where Fig. 11 illustrates failures induced by visual semantics and Fig. 12 illustrates failures induced by query semantics.

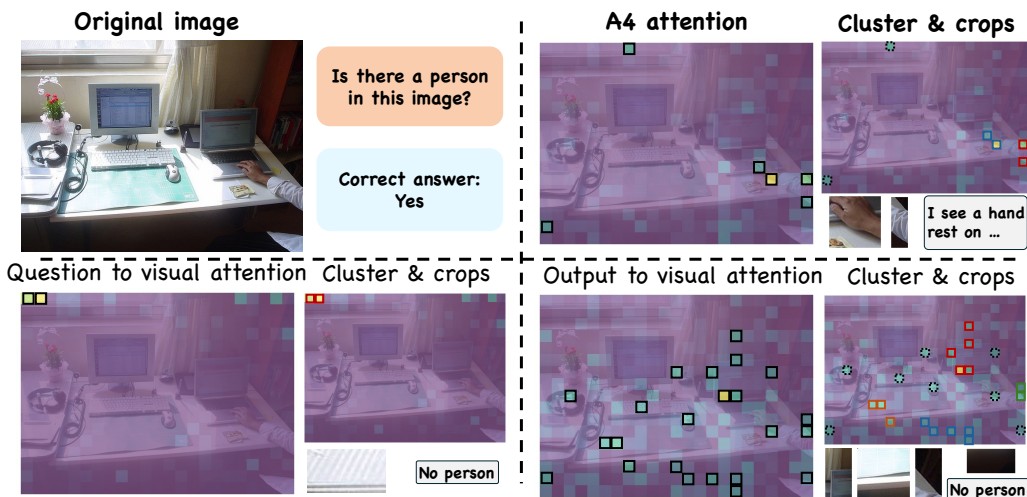

Figure 8: **Case study on attention choices.** A$^4$ attention identifies question-relevant tokens and correctly crops the hand, thereby guiding the model to the correct answer. *Question to visual* denotes cross-attention between the input question and input visual tokens; because the model is not trained for this alignment, the signal is weak. *Output to visual* directly leverages generation-side attention, which is mainstream in MLLMs; however, in MLRMs, reasoning drift introduces numerous distractors.

# D  Q2O AND V2O HEATMAPS

**Setup and notation.** We adopt the standard Transformer attention view for decoder-style ML-RMs (Vaswani et al., 2017): at output step $t$, the model attends over the input-side question tokens $\mathbf{X}_q = \{x_{q,i}\}_{i=1}^{N_q}$ and visual tokens $\mathbf{X}_v = \{x_{v,j}\}_{j=1}^{N_v}$, producing head-aggregated, key-normalized weights $A_{t,i}^{q \to o} \in [0,1]$ and $A_{t,j}^{v \to o} \in [0,1]$ with $\sum_i A_{t,i}^{q \to o} = 1$ and $\sum_j A_{t,j}^{v \to o} = 1$. Unless otherwise noted we average across layers/heads (a common practice in attention-based attribution *post hoc* analyses). COCO categories and bounding boxes $(x, y, w, h)$ are denoted by cat and bbox and follow the dataset's instance annotations (Lin et al., 2014).

**q2o heatmap (question → output).** Given an object string cat, we first detect *object mentions* in the generated output $\{y_t\}_{t=1}^T$ by letters-only matching (case-insensitive) with a leniency margin (token length $\leq |\text{cat}|+2$) and POS tag NOUN. Let the set of indices of these mentions be $\mathcal{T}_{\text{obj}}$. For every output token $t$ with POS NOUN, define its distance to the nearest object mention $d(t) = \min_{u \in \mathcal{T}_{\text{obj}}} |t - u|$. We form seven noun buckets $B_d = \{t : \text{POS}(t) = \text{NOUN}, d(t) = d\}$ for $d \in \{0, \ldots, 6\}$, and three POS buckets over all outputs $B_{\text{ADJ}}, B_{\text{VERB}}, B_{\text{OTHER}}$. The $N_q \times 10$ matrix $M$ used for the q2o heatmap is

$$M_{i,k} = \frac{1}{|B_k|} \sum_{t \in B_k} A_{t,i}^{q \to o} \quad \text{for } k \in \{0, \ldots, 6, \text{ADJ}, \text{VERB}, \text{OTHER}\},$$

with empty buckets contributing zeros. We visualize $M$ as a diverging heatmap (high → red, low → blue) with smooth interpolation; the x-axis shows different buckets, and the y-axis lists question tokens.

**v2o heatmap (visual → output around a bbox).** For each COCO bounding box bbox of category cat present in the output, we locate the nearest occurrence index $t_{\text{cat}}$ of cat (letters-only matching; multiword matched as a contiguous phrase). We extract a temporal window $W = \{t_{\text{cat}} - 10, \ldots, t_{\text{cat}} + 10\} \cap [1, T]$ (21 positions) and the corresponding rows of the visual-to-output attention, giving $A_{W,:}^{v \to o} \in \mathbb{R}^{21 \times N_v}$. Using the patch grid $(t_h, t_w)$ from the resized image, we map bbox to its covered patch set $P_{\text{bbox}} \subseteq \{1, \ldots, N_v\}$ by IoU$> 0.5$ between bbox and each patch cell (pixel-aligned rectangles). Let $r_{\min} : r_{\max}$ and $c_{\min} : c_{\max}$ be the row/column spans of $P_{\text{bbox}}$ on the $(t_h \times t_w)$ grid. For lateral offsets $s \in \{-5, \ldots, 5\}$ we define expanded regions

$$P_{\text{bbox},s} = \{(r,c) : r \in [r_{\min}, r_{\max}], \ c \in [c_{\min} + \min(0,s), \ c_{\max} + \max(0,s)]\} \cap [0, t_h - 1] \times [0, t_w - 1],$$

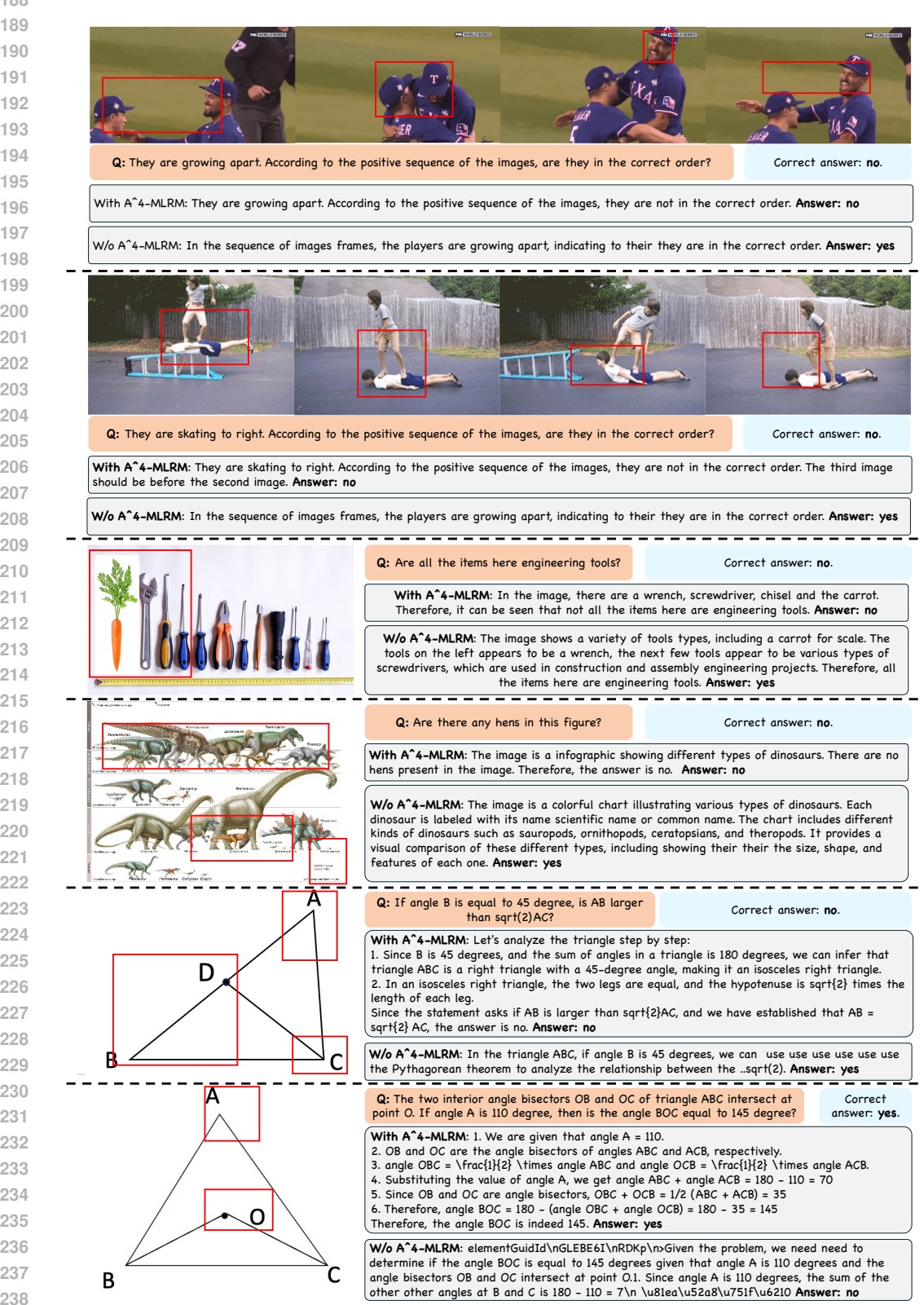

Figure 9: **Case study** (Part One). The visual crops selected by $A^4$-MLRM are highlighted with red bounding boxes in the figure.

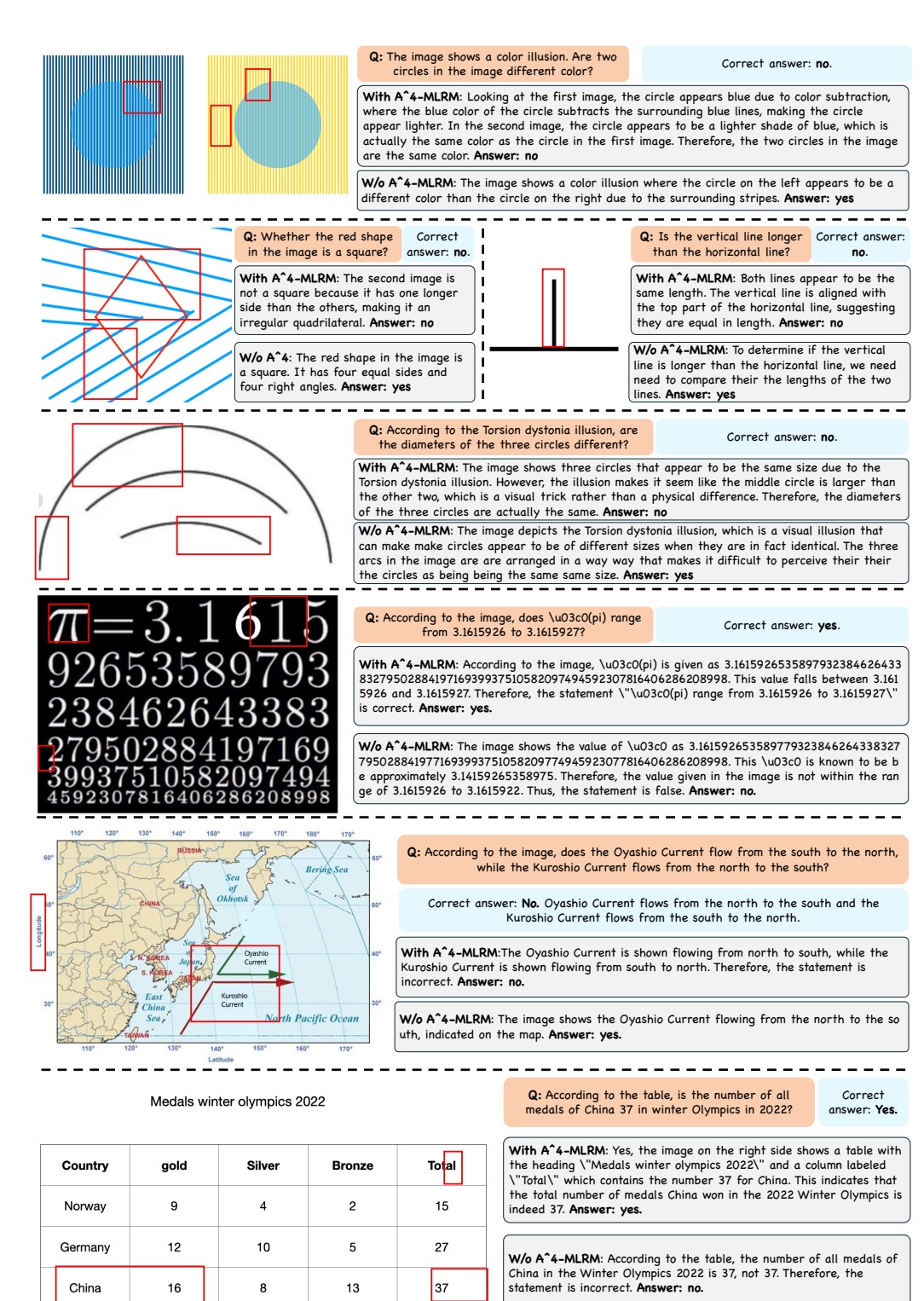

Figure 10: **Case study** (Part Two). The visual crops selected by $A^4$-MLRM are highlighted with red bounding boxes in the figure.

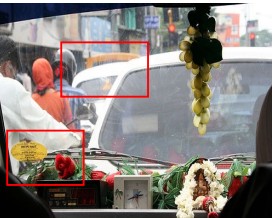

**Q:** Is there a person in the image?  Correct answer: **yes**.

**With A^4-MLRM:** No, there is no person in the image. The image shows a dining table with plates of food, a fork and a knife. There is a white chair and a black bag in the background. The yellow paint on the wall adds a warm touch to the scene. **Answer: no**

**W/o A^4-MLRM:** In the image, there is a woman and two children sitting at a table. The woman appears to be be feeding or assisting one of the children with food while the other child is seated across from her, engaged in eating or playing. **Answer: yes**

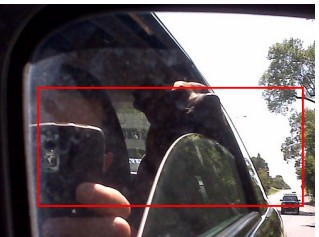

**Q:** Is there a truck in the image?  Correct answer: **no**.

**With A^4-MLRM:** This image seems to be taken from inside the passenger seat of a vehicle, showing the view through the windshield. The windshield has decorations, including flowers hanging from the top. Outside the windshield, there are other vehicles. The truck is partially visible, with its front end and part of the side showing. It's a large vehicle with a boxier shape, which is characteristic of trucks. **Answer: yes**

**W/o A^4-MLRM:** The image shows a view from inside a vehicle, likely a car, looking at another vehicle outside. The other. The vehicle in the background appears to be a truck, as indicated as it has a boxy. **Answer: no**

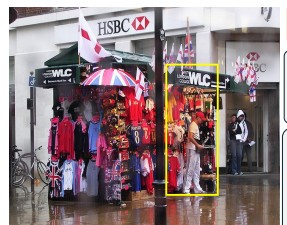

**Q:** Is there a truck in the image?  Correct answer: **yes**.

**With A^4-MLRM:** The image shows a person taking a picture from the inside of a car, with a dog in the back seat looking out the window. The background features a road with trees and a clear sky.. The road is relatively straight and there are no other vehicles visible in the image. **Answer: no**

**W/o A^4-MLRM:** In the image, there appears to be be a reflection and views outside. The reflection of shows the inside of the vehicle where someone's hand, holding a phone, can is reflecting in the driver side.'s side view mirror. The outside view in the side right side of shows vehicles driving on a road with trees lining the side. However, the image is not clear enough to distinguish specific features like the shape or features of a truck. **Answer: yes**

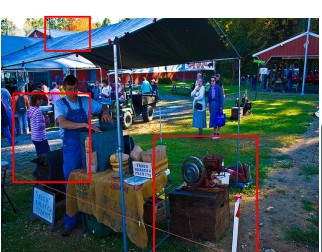

**Q:** Is there a hadbag in the image?  Correct answer: **no**.

**With A^4-MLRM:** There is a person standing near the storefront with a handbag hanging from their shoulder. This can be inferred from the typical way people carry handbags when shopping or walking, and the position of the person relative to the display. **Answer: yes**

**W/o A^4-MLRM:** The image shows a street vendor's stall with various items for sale, including flags and sports attire. The vendor appears to be a man standing behind the stall, and there are a few people people standing in front of the stall. The stall is located outside the HSBC bank building, and the wet street suggests recent rain. Given this information, it is likely that there are a handbag in the image, although it is not explicitly visible, but this can be a reasonable assumption. **Answer: no**

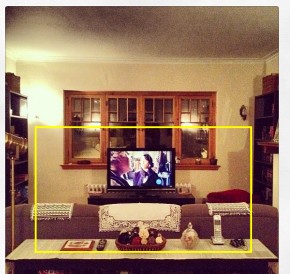

**Q:** Is there a backpack in the image?  Correct answer: **yes**.

**With A^4-MLRM:** No, there is no backpack in the image. The image shows a market stall with a man operating a machine that roasts peanuts. There are a few people in the background, and some boxes and crates are visible. There is no visible backpack in the image. **Answer: no**

**W/o A^4-MLRM:** The image shows an outdoor market scene with several people gathered under tents. In the foreground, there is a man in overalls standing behind a table with various items, including a sign reading \"TRESH ROASTED PEANUts\". There is a small green cooler to the left of the table, and a wooden box to the right. The table has bags of peanuts, and other another box and some some other a small cooler to left in the table, and a wooden box to the ri. Some items including the right. In the background, there there are people are some are some people a small cooler to the left of the table, and a wooden box to the right. Some items to the right. There is a person in the right background holding what appears to be a baby, and some others standing and talking. The grass is green, and there the background, there there there **Answer: yes**

**Q:** Is there a person in the image?  Correct answer: **yes**.

**With A^4-MLRM:** No, there is no person in the image. The images appear to be identical, showing a living room with a couch, a table in front of it, and a television displaying a scene. The image is a close-up of a textured surface that does not contain a person. Therefore, the presence of a person cannot be determined from the image. **Answer: no**

**W/o A^4-MLRM:** The image shows a living room with a television that appears to be displaying a scene with people a person is on the screen. Therefore, there is a person in the image. **Answer: yes**

Figure 11: **Failure case study** (Part One). The visual crops selected by $A^4$-MLRM are highlighted with red and yellow bounding boxes. We observe that, in most cases, $A^4$-MLRM crops the correct regions, but it fails to do so in the third example.

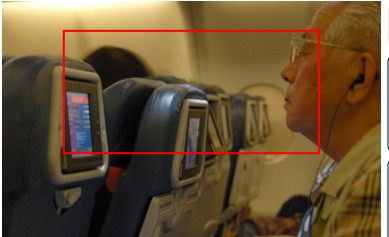

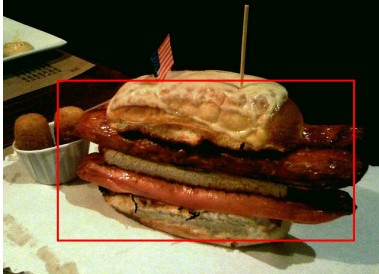

**Q:** Is there a cup in the image?

Correct answer: **no**.

**With A^4-MLRM**: Yes, there is a cup in the image. The image shows a glass of beer on the table, which can be considered a cup. Therefore, there is a cup in the image. **Answer: yes**

**W/o A^4-MLRM**: The image shows a man sitting at a table with a variety of items. There are a plate of food in front of him, including a bottle of beer to his left, and some two glasses of what appears to be beer. He is holding a smartphone in his hand, and appears to be be taking a picture of his food. The glasses on the table have a shape and structure consistent with wine glasses, which are are typically used for beer servings, but cannot be confirmed as such without further context. Therefore, it is not possible to definitively conclude that the glasses are contain beer, but they are be suggestive of that be beer. **Answer: no**

**Q:** Is there a vase in the image?

Correct answer: **no**.

**With A^4-MLRM**: Yes, there is a vase visible in the image. It is located on a white surface, likely a shelf or countertop, and contains a yellow flower arrangement. The vase has a rectangular shape and is positioned in the lower right corner of the image. The white surface it's on appears to be part of a bathroom setting, with tiles visible in the background. The overall scene suggests a clean and organized space, with the vase and flower adding a touch of\u81ea\u7136 (natural) beauty to the room. **Answer: yes**

**W/o A^4-MLRM**: The image shows a bathroom with a toilet, a sink, and a bathtub. The bathroom is well-l-l\n<translation>\n **Answer: no**

**Q:** Is there a chair in the image?

Correct answer: **yes**.

**With A^4-MLRM**: The image shows a row of seats on an airplane, with a man sitting in one of them. The seats have personal entertainment systems on the backrests, which is a common feature on planes. The man appears to be using the in-flight entertainment system, possibly playing a video or watching a movie. There is no chair in this image. **Answer: no**

**W/o A^4-MLRM**: The image shows the interior of an airplane, focusing on the row of seats with personal entertainment systems embedded in the headrests. The seats are are clearly visible, with their backrests and\u5934 the screens in the front of each seat. This man is seated in one of these seats, watching looking ahead. Therefore, it is reasonable to conclude that the image does indeed contain chairs. **Answer: yes**

**Q:** Is there a dog in the image?

Correct answer: **yes**.

**With A^4-MLRM**: The image contains food of some sort, possibly a sandwich or burger, and there is a small flag visible on top of it. Upon further analysis, it appears that the food item is a hot dog, which is a type of sandwich. Therefore, there is no visiable dog in the image. **Answer: no**

**W/o A^4-MLRM**: The image shows a large sandwich with various ingredients, including a hot dog. The top part of the sandwich appears to be a hot dog bun with a slice of cheese and possibly a sausage. the bottom part is another slice of bread, and there the sausage appears to be the main component of the sandwich. Therefore, the image does contain a dog, which is the sausage. **Answer: yes**

Figure 12: **Failure case study** (Part Two). The visual crops selected by $A^4$-MLRM are highlighted with red/yellow boxes. We observe that these failures are driven by object-level semantics: in these examples, the model recognizes semantically similar but incorrect concepts (e.g., *glass* vs. *cup*, *seat* vs. *chair*).

flattened to patch indices. The $21 \times 11$ matrix $V$ for `bbox` is

$$V_{r,\,s+5} = \frac{1}{|P_{\mathrm{bbox},s}|} \sum_{j \in P_{\mathrm{bbox},s}} A^{v \to o}_{t_r,\,j}, \qquad t_r \in W,$$

summarizing how attention over visual patches varies near the bounding box versus its lateral background. We render $V$ with the same diverging colormap and smooth interpolation; the y-axis uses three ticks (others–{cat}–others) centered at $t_{\mathrm{cat}}$, and the x-axis uses three ticks (background–bbox patches–background) centered at $s=0$.

# E  SENSITIVITY, SCALABILITY, AND DEPLOYMENT COST ANALYSIS

We further analyze the sensitivity of $A^4$-MLRM to the thresholds $\tau_q$, $\tau_o$, $\tau_v$ and the clustering parameter dbscan-$\epsilon$. As shown in Tables 8–11. We also summarize latency and throughput in Table 12, and report layer-wise AttnRecall on larger 32B models and LLaVA architecture (Liu et al., 2023a) in Table 13 and Table 14.

Table 8: Sensitivity of accuracy to $\tau_q$ on HallusionBench and POPE.

| Method | 0.5 | 1 | 1.5 | 2 | 3 |
|---|---|---|---|---|---|
| HallusionBench + MM-Eureka | 63.51 | 65.10 | 63.51 | 63.15 | 61.65 |
| POPE + MM-Eureka | 79.86 | 81.38 | 80.32 | 79.13 | 80.92 |
| POPE + Orsta | 83.32 | 83.67 | 85.01 | 83.12 | 82.91 |

Table 9: Sensitivity of accuracy to $\tau_o$ on HallusionBench and POPE.

| Method | 0.5 | 1 | 1.5 | 2 | 3 |
|---|---|---|---|---|---|
| HallusionBench + MM-Eureka | 66.43 | 65.10 | 63.86 | 63.42 | 63.77 |
| POPE + MM-Eureka | 82.32 | 81.38 | 79.22 | 79.62 | 80.62 |
| POPE + Orsta | 84.42 | 83.67 | 85.02 | 83.12 | 79.02 |

Table 10: Sensitivity of accuracy to $\tau_v$ on HallusionBench and POPE.

| Method | 0.5 | 1 | 1.5 | 2 | 3 |
|---|---|---|---|---|---|
| HallusionBench + MM-Eureka | 64.22 | 65.46 | 65.10 | 63.95 | 63.45 |
| POPE + MM-Eureka | 82.92 | 82.41 | 81.38 | 80.92 | 79.12 |
| POPE + Orsta | 81.82 | 83.22 | 83.67 | 85.81 | 83.22 |

Table 11: Sensitivity of accuracy to DBSCAN-$\epsilon$ on HallusionBench and POPE.

| Method | 1.1 | 1.9 | 2.1 | 2.9 | 3.1 | 3.9 |
|---|---|---|---|---|---|---|
| HallusionBench + MM-Eureka | 63.31 | 64.57 | 64.48 | 65.10 | 65.20 | 65.37 |
| POPE + MM-Eureka | 78.52 | 79.62 | 80.62 | 81.38 | 81.32 | 81.22 |
| POPE + Orsta | 84.52 | 84.62 | 85.41 | 83.67 | 84.52 | 83.02 |

**Sensitivity Analysis.** As summarized in Tables 8–11, the accuracies vary only mildly across a wide range of $\tau_q$, $\tau_o$, $\tau_v$, and dbscan-$\epsilon$, indicating that $A^4$-MLRM is not overly sensitive to these hyperparameters. In all main experiments, we therefore fix $\tau_q = 1$, $\tau_o = 1$, $\tau_v = 1.5$, and dbscan-$\epsilon = 2.9$.

**Deployment Cost Analysis.** We conduct a deployment cost analysis of $A^4$-MLRM and report end-to-end costs under different Stage 1 output lengths. All experiments are run with a 7B model on a single NVIDIA GeForce RTX 3090 (20G). As shown in Table 12, longer Stage 1 sequences provide richer priors to Stage 2 but also incur higher inference cost, which is consistent with our expectation.

Table 12: Latency and throughput under different online and offline settings.

| Mode / sequence length | offline / – | online / 1 | online / 20 | online / 50 | online / 128 | online / 256 |
|---|---|---|---|---|---|---|
| Latency (hours / 1000 items) | 0.91 | 0.97 | 1.08 | 1.36 | 1.62 | 1.69 |
| Throughput (tokens/s) | 39.80 | 35.52 | 31.24 | 27.71 | 20.91 | 19.14 |

**Scalability Analysis.** As shown in table 13 and 14, on both 32B variants and models with different underlying architectures, the layer-wise ATTNRECALL exhibits the same rise then fall pattern as in 7B models, and the peak perceptual layers still lie in the later part of the network. For both Orsta-32B and MM Eureka-32B, the global maximum of ATTNRECALL appears around layer 40 out of 64, which corresponds to roughly 70% of the total depth. This indicates that our perception layer finding scales to larger parameter regimes.

Table 13: Layer-wise ATTNRECALL on 32B models.

| Model | 0 | 12 | 24 | 30 | 36 | **40** | 42 | 48 | 54 | 63 |
|---|---|---|---|---|---|---|---|---|---|---|
| MM-Eureka-32B | 0.3060 | 0.3377 | 0.4978 | 0.5358 | 0.5781 | **0.6271** | 0.5474 | 0.4854 | 0.3992 | 0.3099 |
| Orsta-32B | 0.3165 | 0.3397 | 0.4922 | 0.5381 | 0.5669 | **0.6350** | 0.5596 | 0.4961 | 0.4059 | 0.3172 |

Table 14: Layer-wise ATTNRECALL on LLaVA-1.5-7B and LLaVA-Mistral-7B.

| model | arch | 0 | 6 | 12 | 17 | 18 | 24 | 31 |
|---|---|---|---|---|---|---|---|---|
| LLaVA-1.5-7B | LLaVA-1.5 | 33.81% | 38.00% | 43.04% | **50.59%** | 48.29% | 47.03% | 39.02% |
| LLaVA-Mistral-7B | LLaVA-NeXT | 10.37% | 23.95% | 46.00% | 49.04% | **49.72%** | 41.10% | 35.95% |

# F LLM USAGE

We used a large language model **[ChatGPT 5]** strictly for writing assistance limited to spelling, grammar, and minor stylistic polishing.

