# OpenReview forum: "$A^{4}$-MLRM: Fourfold Attention for Adaptive Hallucination Suppression in Multimodal Large Reasoning Model"
_ICLR.cc/2026/Conference — Submitted to ICLR 2026_

### Official Review · Reviewer_htgT · 2025-10-27

**Soundness:** 2
**Presentation:** 1
**Contribution:** 2
**Rating:** 4
**Confidence:** 3

**Summary:**

The paper introduces A4-MLRM, an architecture-agnostic, training-free, and parameter-free plugin designed to mitigate hallucination and reasoning drift (i.e., attentional diffusion toward task-irrelevant details) in MLRMs. They leverage the model’s native attention mechanism, and their strategy results in an average improvement of 1.21× on reasoning benchmarks and 1.16× when transferred to non-reasoning settings.

**Strengths:**

- A4-MLRM is completely training-free and parameter-free, resulting in minimal deployment cost, a significant advantage over prior training-stage mitigation efforts; it's also architecture-agnostic.

- A4-MLRM achieves substantial performance gains, demonstrably improves all reasoning models tested (R1-OneVision, Ocean-R1, MM-Eureka, ORSTA-R1). It also successfully transfers to non-reasoning MLLMs (LLaVA-1.6-Mistral and Qwen2.5-VL), moving some models "from near chance to the GPT-4V range" on perception benchmarks.

**Weaknesses:**

- The paper suffers from several presentation issues. It is difficult to follow, as many notations are introduced at different stages without a consistent framework established in the Background section. Moreover, key steps and observations are only presented in the Appendix, which makes the paper harder to read and the findings more difficult to assess and trust. The figures, rather than clarifying the content, are quite dense and make the paper harder to follow. I would recommend a thorough revision of **the structure of this paper**, perhaps by emphasizing the most important observations or focusing more clearly on illustrating the method itself.

- A4-MLRM currently relies on a two-stage inference pipeline (Stage 1: attention mining; Stage 2: focused re-inference), which, given current reasoning-model architectures, may introduce latency and computational overhead. Moreover, the paper lacks a clear comparison of computational costs and how they scale with output length. The accuracy of online inference is closely tied to the output sequence length used in Stage 1: longer sequences provide richer priors for Stage 2 and thus higher accuracy, highlighting a trade-off between efficiency and performance, I guess...

**Questions:**

Q1: The paper identifies that the perception signal peaks around layers 18–24 in 7B architectures using AttnRecall. How stable is this finding across different model sizes (e.g., 13B or 72B variants mentioned in related work or baselines like Qwen2.5-VL) or models with fundamentally different underlying architectures?

Q2: For Ocean-R1, in the case of POPE (Table 5), we observe a slight decrease in accuracy after applying A4-MLRM (from 86.77% accuracy without A4 to 85.76% with A4). Although it is quite minor, could the authors suggest reasons for such behavior? Could A4-MLRM, in some cases, exclude relevant visual context necessary for certain complex reasoning or general QA tasks?

---

> ### Author Response · Authors · 2025-11-20
> **[1/2] Author response**
>
> We would like to express our deepest respect for your review! In response to your comments, we have carefully prepared a point-by-point reply:
> >**W1:** The paper suffers from several presentation issues. I would recommend a thorough revision of **the structure of this paper**, perhaps by emphasizing the most important observations or focusing more clearly on illustrating the method itself.
>
> We thank the reviewer for the helpful suggestions on improving the presentation and structure of the paper. We have **thoroughly revised** the manuscript with the following changes:
> 1. **Unified notation in main text.** To improve readability and consistency, we add a unified notation table in the Preliminaries section (Table 1) and ensure that all symbols introduced later follow this framework.
> 2. **Observations as a standalone section.** We reorganize the observations into a dedicated section in the main paper, where we clearly present and discuss each key observation.
> 3. **Refactored Method section.** We substantially rewrite the Method section to improve clarity. In this revision, we remove redundant symbols (for example, the original $h$, $g$, $U$, $B$) and unify the notation by using $X$ for inputs, $y$ for outputs, and $A$ for attention. We also add explanatory text to each subsection to walk through the method itself, so that the workflow is easier to follow.
> 4. **Clarified figures focusing on the method.** We simplify the model overview figure (Figure 3) and related plots to focus on illustrating the method itself and its main information flow, rather than low-level implementation details.
>
> We hope these revisions make the structure more coherent and the overall paper easier to follow, and we sincerely look forward to your feedback.
>
> ---
>
> >**W2:** A4-MLRM currently relies on a two-stage inference pipeline (Stage 1: attention mining; Stage 2: focused re-inference), which, given current reasoning-model architectures, may introduce latency and computational overhead. Moreover, the paper lacks a clear comparison of computational costs and how they scale with output length. The accuracy of online inference is closely tied to the output sequence length used in Stage 1: longer sequences provide richer priors for Stage 2 and thus higher accuracy, highlighting a trade-off between efficiency and performance, I guess...
>
> We completely agree with your observation about the two-stage inference pipeline. Our experimental results lead to the **same conclusion** (line 474 in the manuscript), namely that “a longer Stage 1 output provides richer priors to Stage 2, thus yielding higher accuracy.”
>
> We also fully understand the concern about computational costs. We therefore report end-to-end throughput under different Stage 1 output lengths. All experiments are conducted with a 7B model on a single NVIDIA GeForce RTX 3090 (20G). The results are summarized below:
>
> |mode / sequence length|offline / –|online / 1|online / 20|online / 50| online / 128|online / 256|
> |-|-|-|-|-|-|-|
> |latency (hours / 1000 items)|0.91|0.97|1.08|1.36|1.62|1.69|
> |throughput (tokens/s)| 39.80| 35.52|31.24|27.71|20.91|19.14|
>
> In line with prior inference-time hallucination mitigation methods based on general MLLMs [1-2], our approach also relies on multiple forward passes.
>
> ---
>
> > **Q1:** The paper identifies that the perception signal peaks around layers 18–24 in 7B architectures using AttnRecall. How stable is this finding across different model sizes or models with fundamentally different underlying architectures?
>
> We thank the reviewer for this question and we fully understand the concern about the generality of the AttnRecall based layer analysis. To address this, we compute layer wise AttnRecall on models with different architectures and parameter scales, including three 32B VLMs and additional 7B variants, as summarized below.
>
> *32B models*
> | |0|12|24|30|36|**40**|42|48|54|63|
> |-|-|-|-|-|-|-|-|-|-|-|
> |mm-eureka-32b|0.3060|0.3377|0.4978|0.5358|0.5781|**0.6271**|0.5474|0.4854|0.3992|0.3099|
> | orsta-32b|0.3165|0.3397|0.4922|0.5381|0.5669|**0.6350**|0.5596|0.4961|0.4059|0.3172|
> | qwen2.5-vl-32b|0.2998|0.3376|0.5026|0.5394|0.5779|**0.6482**|0.5509|0.5061|0.4121|0.3044|
>
> *More 7B models*
>
> | |0|6|12|18|24|27|
> |-|-|-|-|-|-|-|
> |cof-7b [3]| 0.2848|0.2861|0.4377|**0.5340**|0.4530|0.4104|
> |M2-Reasoning-7b [4]|0.3066|0.3118|0.4280|0.4855|**0.5416**|0.4629|
>
> Across all these models, AttnRecall consistently shows **a rise then fall pattern** across depth, and the peak perceptual layers appear in the later part of the network, roughly around **70% of the total depth**. In particular, all three 32B models reach their maximum AttnRecall around layer 40 out of 64, and the 7B models also peak in the later layers. These results suggest that our observation about the concentration of perception signals in later layers is stable across different model sizes and underlying architectures, including larger variants such as Qwen2.5-VL.

---

> ### Author Response · Authors · 2025-11-20
> **[2/2] Author response**
>
> > **Q2:** For Ocean-R1, in the case of POPE (Table 5), we observe a slight decrease in accuracy after applying A4-MLRM (from 86.77% accuracy without A4 to 85.76% with A4). Although it is quite minor, could the authors suggest reasons for such behavior? Could A4-MLRM, in some cases, exclude relevant visual context necessary for certain complex reasoning or general QA tasks?
>
>
> We thank the reviewer for this careful observation and, to address this issue, we analyze the behavior of Ocean-R1 on POPE in detail. Specifically, we focus on the 304 cases where the original model (w/o A4) is correct but the model with $A^4$-MLRM is wrong, which corresponds to 3.38\% of the full dataset. We manually categorize these cases into three types:
>
> - **Unfaithful**: the reasoning process is correct, but the final conclusion is wrong, which is a common defect of large models [5-7]. This type accounts for **45.1%** of these errors.
> - **A4-failure**: $A^4$-MLRM routes attention to incorrect regions, which leads the model to a wrong prediction while the original model is correct. This type accounts for **4.3%** of these errors.
> - **Semantic**: $A^4$-MLRM guides attention to the correct regions, but the model still makes a wrong decision due to visual or textual semantic uncertainty (for example, “Is a hot dog a dog” or “Is a flowerpot a vase”). This type accounts for **50.6%** of these errors.
>
> The errors **directly** caused by $A^4$-MLRM are very small in absolute terms, accounting for only about $ 3.38\\% \times 4.3\\% \approx 0.15\\% $ of the full dataset. Most of the remaining cases are due to model **unfaithfulness**, which is a well known phenomenon and can be more noticeable for models that are not sufficiently trained on multi image setups [6,7], or to **visual or textual semantics** that lead the model to incorrect judgments. We agree that there exist rare cases where $A^4$-MLRM may exclude relevant visual context that is necessary for certain tasks, but given their very **low frequency**, the method still brings consistent improvements in most settings.
>
> To further demonstrate the behavior on more complex and realistic tasks, we also evaluate on a multi-object spatial reasoning dataset 3DSRBench [8] and a real-world QA dataset RWQA [9]:
>
> | data      | Ocean-r1 | with A4 | Orsta  | with A4 | MM-Eureka | with A4 |
> | --------- | -------- | ------- | ------ | ------- | --------- | ------- |
> | 3DSRBench [8] | 37.66%   | 40.06%  | 43.44% | 45.14%  | 56.25%    | 59.24%  |
> | RWQA [9]     | 63.27%   | 70.33%  | 65.23% | 67.84%  | 60.39%    | 65.49%  |
>
> These results show that $A^4$-MLRM broadly improves performance on complex reasoning and real-world understanding tasks, while the proportion of cases where it excludes relevant context remains very small.
>
> ## References
>
> [1] Leng, Sicong, et al. "Mitigating object hallucinations in large vision-language models through visual contrastive decoding." CVPR 2024.
>
> [2] Li, Zhuowei, et al. "The hidden life of tokens: Reducing hallucination of large vision-language models via visual information steering." ICML 2025.
>
> [3] Zhang, Xintong, et al. "Chain-of-Focus: Adaptive Visual Search and Zooming for Multimodal Reasoning via RL." arXiv preprint arXiv:2505.15436 (2025).
>
> [4] AI, Inclusion, et al. "M2-reasoning: Empowering mllms with unified general and spatial reasoning." arXiv preprint arXiv:2507.08306 (2025).
>
> [5] Turpin, Miles, et al. "Language models don't always say what they think: Unfaithful explanations in chain-of-thought prompting." NeurIPS 2023.
>
> [6] Arcuschin, Iván, et al. "Chain-of-thought reasoning in the wild is not always faithful." arXiv preprint arXiv:2503.08679 (2025).
>
> [7] Chen, Yanda, et al. "Reasoning Models Don't Always Say What They Think." arXiv preprint arXiv:2505.05410 (2025).
>
> [8] Ma, Wufei, et al. "3dsrbench: A comprehensive 3d spatial reasoning benchmark." CVPR 2025.
>
> [9] https://huggingface.co/datasets/visheratin/realworldqa

---

> ### Author Response · Authors · 2025-11-24
> **Further validation of AttnRecall stability across architectures (Q1)**
>
> We thank you again for raising Question 1. To further address this question, we have additionally evaluated AttnRecall on two models with fundamentally different architectures, namely **LLaVA-1.5 and LLaVA-Mistral**. The layer-wise results are summarized below:
>
> | model          | architecture    | 0 | 6 | 12 |17 |18 | 24 | 31 |
> |----------------|-------------|---|---|----|----|-|----|----|
> | LLaVA-1.5-7b      | LLaVA-1.5   | 33.81% |38.00% | 43.04% |**50.59%** | 48.29% |47.03% | 39.02% |
> | LLaVA-mistral-7b  | LLaVA-NeXT  | 10.37%  | 23.95%|46.00%   | 49.04%| **49.72%** | 41.10% | 34.95% |
>
> Across these different architectures, the perception capability exhibits a consistent rise–then–fall pattern, with the peak appearing at around 70% of the total depth. We have incorporated these new results into the latest version of the manuscript and look forward to any further comments from the reviewer.

---

> > ### Comment · Area_Chair_iQeH · 2025-11-26
> >
> > Dear Reviewer,
> >
> > Thanks for your time and effort in reviewing ICLR2026 submissions. The authors have provided their responses to your reviews. Please read and raise your further comments, and discuss with the authors.
> >
> > Best regards,
> >
> > Your AC

---

### Official Review · Reviewer_XuFa · 2025-10-30

**Soundness:** 3
**Presentation:** 2
**Contribution:** 2
**Rating:** 4
**Confidence:** 4

**Summary:**

This paper introduces A4-MLRM, a training-free and architecture-agnostic inference-time approach to mitigate hallucinations in multimodal reasoning models. The method traces attention from question tokens to generated reasoning tokens and subsequently to visual patches, and re-queries the model using attention-selected regions. Experiments show improvements across several reasoning models and hallucination benchmarks.

**Strengths:**

- Addresses a challenge: hallucination in multimodal reasoning models.
- Practical and deployment-friendly design: no training, no architectural modifications.
- Demonstrates consistent improvements across multiple MLLMs and evaluation benchmarks.
- Includes attention layer analysis and ablations supporting empirical choices.
- Shows transferability to non-reasoning models, indicating broader applicability.

**Weaknesses:**

- Innovation is limited; the idea resembles prior attention-guided focusing / grounding strategies and is largely heuristic.
- Heavy reliance on attention as a meaningful signal without deeper theoretical justification on its causal reliability.
- Evaluation focuses mainly on binary hallucination settings; generalization to open-ended reasoning and complex visual tasks is unclear.
- Sensitivity to thresholds, clustering, and sequence length is not systematically studied.
- Two-stage inference incurs latency, and deployment cost analysis is insufficient.
- Behavior on larger models (>7B) is not explored, raising concerns about scalability.

**Questions:**

1. How reliable is attention for grounding when attention maps are known to be noisy or misaligned in some models?
2. Are τq, τo, τv and clustering parameters fixed across all models and datasets? Any sensitivity analysis?
3. How does the method perform on open-ended VQA and compositional reasoning tasks?
4. Can you report latency and compute overhead for both online and offline modes?
5. Does performance scale or saturate on larger models (e.g., 34B / 70B)?
6. Can you provide failure case visualizations where attention routing misleads the model?
7. How does the method compare or combine with RL-based grounding or verifier-based hallucination mitigation?
8. Does the method risk over-focusing on small regions, losing necessary context for multi-entity reasoning?
9. Recent work suggests that extended chain-of-thought can weaken visual grounding and increase hallucinations, closely related to the “reasoning drift” discussed here. Could the authors clarify how A4-MLRM relates to these findings and whether it mitigates the trade-off between deeper reasoning and degraded perception, especially in long-chain reasoning scenarios?

- More Thinking, Less Seeing? Assessing Amplified Hallucination in Multimodal Reasoning Models
-  More Thought, Less Accuracy? On the Dual Nature of Reasoning in Vision-Language Models

---

> ### Author Response · Authors · 2025-11-20
> **[1/4] Author response**
>
> We sincerely thank you for the thoughtful and constructive reviews of our manuscript! Based on your questions and recommendations, we give point-by-point responses to your comments and describe the revisions we made to address them.
> > **W2:** Heavy reliance on attention as a meaningful signal without deeper theoretical justification on its causal reliability.
> **Q1:** How reliable is attention for grounding when attention maps are known to be noisy or misaligned in some models?
>
> We appreciate the reviewer’s concern about treating attention as a meaningful signal. In our work, we do not assume that attention provides a perfect causal explanation. Instead, we use attention as an approximate grounding signal and **explicitly validate** its reliability both qualitatively and quantitatively.
>
> First, to support the basic reliability of attention for visual grounding, we provide extensive qualitative results in Figure 1(b) and quantitative results in Figure 2(b) showing that the model’s attention consistently concentrates on the objects mentioned in the generated descriptions. Recent independent studies [1,2] also report that VLMs often attend to correct visual evidence even when producing incorrect answers, which suggests that attention still encodes meaningful perceptual signals even if the final reasoning sometimes fails[2]. This is aligned with our empirical observations.
>
> Second, we agree that raw attention can contain noise. $A^4$-MLRM **reduces the impact of such noise** through two design choices:
>
> 1. **Layer-wise averaging**
>
>   Consistent with prior findings that different layers specialize in perception and reasoning [3,4], our analysis reveals a similar pattern, and we average attention across heads within each layer. In practice, we observe that single-head attention maps can be noisy, while layer-averaged maps highlight object-level regions more stably. Concretely, on layer $L_{\max}$, we measure the variance of attention in out-of-region (OOR) areas:
>    - head-wise OOR variance: $13.63 \times 10^{-6}$
>    - layer-wise OOR variance: $1.65 \times 10^{-6}$
>
>    This corresponds to roughly an **8× reduction** in OOR variance, indicating that layer-wise aggregation significantly suppresses noise in irrelevant regions and yields more reliable grounding signals.
>
> 2. **DBSCAN-based clustering**
>
>    We adopt DBSCAN as the clustering algorithm exactly because of its **robustness to noise**. When applied to the attention map on the patch grid, DBSCAN treats isolated high-attention pixels as outliers instead of forming clusters, while preserving dense, spatially coherent regions. As analyzed in Appendix C.2, this behavior allows us to focus on consistent attention regions that correspond to objects, while discarding scattered noise.
>
> In summary, our qualitative and quantitative evidence, combined with recent findings in the literature, support that attention carries stable perceptual signals. $A^4$-MLRM is explicitly designed to denoise and aggregate these signals, which makes AttnRecall a reasonably reliable and robust grounding metric in practice.
>
> ---
>
> >**W3&Q3:** Performance on open-ended VQA and compositional reasoning tasks.
>
> We fully understand your concern about the performance of $A^4$-MLRM on open-ended VQA and compositional reasoning tasks. To address this, we evaluate $A^4$-MLRM on the open-ended subset of RH-Bench [5] to test its open-ended capability, and on one compositional reasoning benchmark, 3DSRBench [6] (spatial multi-object reasoning). For open-ended questions, we follow the RH-Bench setting and use Qwen-2.5-VL-32B as the judge model, scoring each answer along hallucination score (0–5, where 0 means fully hallucinated and 5 means no hallucination).
>
> The RH-Bench open-ended results are summarized as follows:
>
> |model| Orsta | Orsta+$A^4$-MLRM | Ocean-r1|Ocean-r1+$A^4$-MLRM |
> |-|-|-|-|-|
> |hallu-score|2.52|**2.65**|2.38|**2.56**|
>
> Results on 3DSRBench are:
>
> |data|Ocean-r1| + $A^4$-MLRM |Orsta| + $A^4$-MLRM |MM-Eureka| + $A^4$-MLRM |
> |-|-|-|-|-|-|-|
> | 3DSRBench |37.66%|**40.06%**|43.44%|**45.14%**|56.25%|**59.24%**|
>
> These results show that on both open-ended and compositional reasoning datasets, $A^4$-MLRM consistently brings stable improvements to the underlying reasoning models.
>
> ---
>
> >**W5&Q4:** Can you report latency and compute overhead for both online and offline modes?
>
> We appreciate your question about the latency and compute overhead. Similar to existing methods during inference [7,8], our approach also relies on multiple forward passes. We report the latency and throughput as follows, all experiments are conducted with a 7B model on a single NVIDIA GeForce RTX 3090 (20G):
>
> |mode / sequence length|offline / –|online / 1|online / 20| online / 50 |online / 128|online / 256|
> |-|-|-|-|-|-|-|
> |latency (hours/1000 items)|0.91|0.97|1.08|1.36|1.62|1.69|
> |throughput (tokens/s)|39.80|35.52|31.24|27.71|20.91|19.14|
>
> We have updated Appendix Table 12 with these results.

---

> ### Author Response · Authors · 2025-11-20
> **[2/4] Author response**
>
> > **W4&Q2:** Sensitivity analysis on τq, τo, τv and clustering parameters
>
> We thank the reviewer for raising the concern about the sensitivity of our method to the thresholds $\tau_q$, $\tau_o$, $\tau_v$ and the clustering parameters. In the main experiments, all reported results use fixed hyperparameters, with $\tau_q=1$, $\tau_o=1$, $\tau_v=1.5$ and dbscan-$\epsilon=2.9$. To assess robustness, we further conduct a systematic sensitivity study on different models and datasets.
>
> **MM-Eureka on HallusionBench**
>
> |$\tau_q$|0.5|1|1.5|2|3|
> |-|-|-|-|-|-|
> |acc|63.51|65.10|63.51|63.15|61.65|
>
> |$\tau_o$|0.5|1|1.5|2|3|
> |-|-|-|-|-|-|
> |acc|66.43|65.10|63.86|63.42|63.77|
>
> |$\tau_v$|0.5|1|1.5|2|3|
> |-|-|-|-|-|-|
> |acc|64.22|65.46|65.10|63.95|63.45|
>
> |dbscan-$\epsilon$|1.1|1.9|2.1|2.9|3.1|3.9|
> |-|-|-|-|-|-|-
> |acc|63.31|64.57|64.48|65.10|65.20|65.37|
>
> **MM-Eureka on POPE**
>
> |$\tau_q$|0.5|1|1.5|2|3|
> |-|-|-|-|-|-|
> |acc|79.86|81.38|80.32|79.13|80.92|
>
> |$\tau_o$|0.5|1|1.5|2|3|
> |-|-|-|-|-|-|
> |acc|82.32|81.38|79.22|79.62|80.62|
>
> |$\tau_v$|0.5|1|1.5|2|3|
> |-|-|-|-|-|-|
> |acc|82.92|82.41|81.38|80.92|79.12|
>
> |dbscan-$\epsilon$|1.1|1.9|2.1|2.9|3.1|3.9|
> |-|-|-|-|-|-|-
> |acc|78.52|79.62|80.62|81.38|81.32|81.22|
>
> **Orsta on POPE**
>
> |$\tau_q$|0.5|1|1.5|2|3|
> |-|-|-|-|-|-|
> |acc|83.32|83.67|85.01|83.12|82.91|
>
> |$\tau_o$|0.5|1|1.5|2|3|
> |-|-|-|-|-|-|
> |acc|84.42|83.67|85.02|83.12|79.02|
>
> |$\tau_v$|0.5|1|1.5|2|3|
> |-|-|-|-|-|-|
> |acc|81.82|83.22|83.67|85.81|83.22|
>
> |dbscan-$\epsilon$|1.1|1.9|2.1|2.9|3.1|3.9|
> |-|-|-|-|-|-|-
> |acc|84.52|84.62|85.41|83.67|84.52|83.02|
>
> Overall, these results support that our method is **not overly sensitive to specific hyperparameter** choices and can transfer well across models and datasets. This observation is consistent with our earlier findings that **critical tokens and non critical tokens behave distinctly during inference**. We have updated Tables 8–11 in the Appendix with the detailed sensitivity analysis results.
>
> >**W6&Q5:** Performance on larger models (>7B).
>
> We thank the reviewers for asking about the behavior of our method on larger models and whether the performance scales or saturates when moving beyond 7B.
>
> To address this concern, we conduct additional experiments on two reasoning oriented VLMs that have larger 32B variants (Orsta-32b and MM-Eureka-32b). We evaluate both (1) the layer wise AttnRecall and (2) the effect of applying $A^4$-MLRM on Hallusion-Bench.
>
> On Hallusion-Bench, we report qAcc, fAcc, hAcc, and aAcc, denoting the accuracy per question pair, per figure, on hard questions, and the overall average:
>
> | model| qAcc  | fAcc  | hAcc  | aAcc  |
> |-|-|-|-|-|
> | Orsta-32b| 33.41 | 41.78 | 47.45 | 64.49 |
> | with $A^4$-MLRM|38.86 | 45.42 | 52.59 | 67.17 |
> | MM-Eureka-32b  | 27.91 | 42.49 | 48.37 | 61.55 |
> | with $A^4$-MLRM   | 33.55 | 48.95 | 55.79 | 68.68 |
>
> We also compute layer-wise AttnRecall for these 32B models:
>
> | | 0| 12| 24 | 30| 36| **40**| 42| 48     | 54     | 63     |
> |-|-|-|-|-|-|-|-|-|-|-|
> | MM-Eureka-32b | 0.3060 | 0.3377 | 0.4978 | 0.5358 | 0.5781 | **0.6271** | 0.5474 | 0.4854 | 0.3992 | 0.3099 |
> | Orsta-32b     | 0.3165 | 0.3397 | 0.4922 | 0.5381 | 0.5669 | **0.6350** | 0.5596 | 0.4961 | 0.4059 | 0.3172 |
>
> These results lead to two observations.
>
> 1. The layer wise AttnRecall on 32B models exhibits **the same rise then fall pattern** as in 7B models, and the peak perceptual layers still lie in the later part of the network. For both Orsta-32B and MM-Eureka-32B, the global maximum of AttnRecall appears around layer 40 out of 64, which corresponds to roughly **70% of the total depth**. This indicates that our perception layer finding scales to larger parameter regimes.
> 2. Applying $A^4$-MLRM on top of these 32B models yields consistent and sometimes sizeable gains on Hallusion-Bench in all metrics. This shows that the proposed method not only remains effective but can further improve the reasoning robustness of larger models, which supports the **scalability** of our approach beyond 7B.
>
> ---
>
> >**Q6:** Can you provide failure case visualizations where attention routing misleads the model?
>
> We provide failure case visualizations in the anonymous repository and in the updated manuscript (Fig. 11-12). From manual inspection, we observe that clear cases where attention routing misleads the model are relatively rare (**~4.3% of failure cases**). In most remaining failures, the attention routing is correct, but the final answers are still wrong because the image content or the question semantics lead the model to an incorrect judgment.

---

> ### Author Response · Authors · 2025-11-20
> **[3/4] Author response**
>
> >**Q7:** How does the method compare or combine with RL-based grounding or verifier-based hallucination mitigation?
>
> We thank the reviewer for pointing out the connection to RL-based grounding and verifier-based hallucination mitigation. $A^4$-MLRM is designed as a plug in mechanism that operates purely at inference time, so it can be directly applied on top of models that already use RL- or verifier-based training objectives.
>
> To verify this in practice, we evaluate $A^4$-MLRM on Hallusion-Bench by combining it with two representative models:
>
> -  **Chain of Focus (CoF) [9]**, which uses RL to train the model to adaptively output bounding boxes during reasoning and crop the image to zoom into key regions for improved multimodal reasoning.
> -  **Reverse [10]**, which integrates hallucination aware training with online self-verification and employs retrospective resampling during generation to detect, roll back, and rewrite suspicious tokens within a single VLM.
>
> | model|type| w/o A4 MLRM | with A4 MLRM |
> |-|-|-|-|
> | CoF-rl-7b| RL-based| 62.01| **66.09**|
> | Reverse-3b|verifier-based| 54.82| **57.09**|
>
> As shown, applying $A^4$-MLRM on top of both RL-based and verifier-based models leads to consistent accuracy improvements on Hallusion Bench. This demonstrates that our method is complementary to these training-based approaches and can further enhance grounding and hallucination robustness without any additional retraining.
>
> ---
>
> >**Q8:** Does the method risk over-focusing on small regions, losing necessary context for multi-entity reasoning?
>
> We fully understand your concern that $A^4$-MLRM might over-focus on small regions and lose necessary context for multi-entity reasoning. In our response to W3/Q3, we have already reported results on multi-entity reasoning benchmarks, where $A^4$-MLRM improves performance, suggesting that it does not harm multi-entity reasoning.
>
> To further analyze the effect of region size, we conduct an additional study on POPE by grouping entities according to their spatial extent in the image. Here, “\# patches” denotes how many patches the entity occupies, “n” is the number of samples in the range, and “avg\_npatches” is the average patch count per group:
>
> | \# patches   | n | avg\_npatches | MM-Eureka | MM-Eureka + A4 | Orsta|Orsta + A4|
> |-|-|-|-|-|-|-|
> | \[0, 4\)| 696 | 1.3| 45.69%| **48.56%**| 45.98% | **49.14%**|
> | \[4, 16\) | 738 | 8.2| 65.04%|**67.48%**| 63.41% | **70.89%**|
> | \[16, 49\)| 696 | 28.6| 75.29%| **77.73%**| 67.24% | **81.47%**|
> | \[49, 100\)| 501 | 73.7| 75.65%| **80.44%**| 68.66% | **83.83%**|
> | \[100, 225\)| 603 | 158.2| 83.73%| **84.58%**| 76.78% |**82.59%**|
> | \[225, 515\)| 300 | 309.2| 79.00%| **81.67%**| 75.33% |**80.00%**|
>
> We observe that $A^4$-MLRM improves accuracy **even for small regions** (e.g., \[0, 4\) patches) and tends to provide larger gains for larger entities.
>
> ---
>
> >**W1:** Innovation is limited; the idea resembles prior attention-guided focusing / grounding strategies and is largely heuristic.
>
> We thank the reviewer for this comment and for pointing out the connection to prior attention-guided focusing and grounding strategies.
>
> Existing methods [8, 11, 12] typically use attention as a coarse heuristic signal for grounding and then modify the decoding process accordingly. They often rely on the overall amount of attention on regions or tokens to reweight logits or suppress suspicious signals, which mainly targets object hallucination at a relatively coarse granularity.
>
> Our work differs from these approaches in two aspects. First, we conduct a systematic multi-layer evaluation of large VLMs and propose **AttnRecall**. In the motivation section, we perform extensive preliminary experiments and build a **fine-grained**, **token-level** analysis of how the model allocates attention over question, visual input, and output tokens, which yields an integrated view of multimodal reasoning and hallucination. Based on these observations, our design uses attention not only as a heuristic, but as a metric-driven mechanism that can reduce multiple types of hallucination (including both perceptual and reasoning hallucinations).
>
> Second, $A^4$-MLRM is implemented as a **plug-in** module that can be seamlessly applied to arbitrary backbone models. We show that it consistently brings improvements across different model architectures and a wide range of datasets.

---

> ### Author Response · Authors · 2025-11-20
> **[4/4] Author response**
>
> >**Q9:** How A4-MLRM relates to recent findings and whether it mitigates the trade-off between deeper reasoning and degraded perception, especially in long-chain reasoning scenarios?
>
> Recent work [5] finds that as the chain-of-thought becomes longer, models gradually ignore the visual modality and over-focus on the textual modality, which is reflected by attention shifting from vision to text. They explain this behavior at a high level (**modality level**) and use it to analyze the causes of hallucination. Our findings with $A^4$-MLRM are complementary. We study hallucination at a finer granularity (**multimodal token level**) and show that under limited visual attention budget [5], reasoning models still need to allocate part of their attention to irrelevant visual evidence or entities for evidence collection. This further clarifies that current reasoning models are suffering from **severe** hallucination.
>
> In long-chain scenarios, these studies suggest that there is an internal “debate” between perception and reasoning in multimodal reasoning models. $A^4$-MLRM mitigates this by explicitly guiding the model to focus on the most relevant regions during reasoning, which **reduces the perceptual burden and lets the model use more capacity for reasoning**. To support this, we measure the average output **token length** on the “general” and “reason” subsets of VMC-Bench:
>
> | split   | w/o $A^4$ | with $A^4$ | increase (%) |
> |-|-|-|-|
> | general | 134.43 | 136.10  | 1.24%|
> | reason|197.22 |302.88|**53.57%**|
>
> We observe that compared with general datasets, $A^4$-MLRM increases the reasoning length more **significantly** on complex reasoning datasets. This also explains why our gains on reasoning benchmarks are **above the average** improvement over all datasets.
>
> ## References
>
> [1] Bi, Jing, et al. "Unveiling visual perception in language models: An attention head analysis approach." CVPR 2025.
>
> [2] Liu, Zhining, et al. "Seeing but Not Believing: Probing the Disconnect Between Visual Attention and Answer Correctness in VLMs." arXiv preprint arXiv:2510.17771 (2025).
>
> [3] Tao, Mingxu, et al. "Probing Multimodal Large Language Models for Global and Local Semantic Representations." LREC-COLING 2024.
>
> [4] Yu, Zhuoran, et al. "How Multimodal LLMs Solve Image Tasks: A Lens on Visual Grounding, Task Reasoning, and Answer Decoding." COLM 2025.
>
> [5] Liu, Chengzhi, et al. "More Thinking, Less Seeing? Assessing Amplified Hallucination in Multimodal Reasoning Models." arXiv preprint arXiv:2505.21523 (2025).
>
> [6] Ma, Wufei, et al. "3dsrbench: A comprehensive 3d spatial reasoning benchmark." CVPR 2025.
>
> [7] Leng, Sicong, et al. "Mitigating object hallucinations in large vision-language models through visual contrastive decoding." CVPR 2024.
>
> [8] Li, Zhuowei, et al. "The hidden life of tokens: Reducing hallucination of large vision-language models via visual information steering." ICML 2025.
>
> [9]  Zhang, Xintong, et al. "Chain-of-Focus: Adaptive Visual Search and Zooming for Multimodal Reasoning via RL." arXiv preprint arXiv:2505.15436 (2025).
>
> [10] Wu, Tsung-Han, et al. "Generate, but Verify: Reducing Hallucination in Vision-Language Models with Retrospective Resampling." arXiv preprint arXiv:2504.13169 (2025).
>
> [11] Park, Woohyeon, et al. "SECOND: Mitigating Perceptual Hallucination in Vision-Language Models via Selective and Contrastive Decoding." ICML 2025.
>
> [12] Cho, Yeongjae, et al. "Do you keep an eye on what i ask? mitigating multimodal hallucination via attention-guided ensemble decoding." ICLR 2025.

---

> > ### Comment · Area_Chair_iQeH · 2025-11-26
> >
> > Dear Reviewer,
> >
> > Thanks for your time and effort in reviewing ICLR2026 submissions. The authors have provided their responses to your reviews. Please read and raise your further comments, and discuss with the authors.
> >
> > Best regards,
> >
> > Your AC

---

### Official Review · Reviewer_ndkQ · 2025-11-01

**Soundness:** 3
**Presentation:** 3
**Contribution:** 2
**Rating:** 8
**Confidence:** 4

**Summary:**

This paper first analyzes the reasons why VLMs generate hallucinations. Then they introduce a series of methods to evaluate visual perception and locate important visual regions based on the attention score. The proposed method leads to consistent performance improvement across various hallucination benchmarks.

**Strengths:**

- This paper provides very interesting insights about reasoning VLMs tend to produce hallucinations due to attention drift, and proposes a pipeline to locate the important visual region based on the model's attention score.

- A4MLRM effectively suppresses the hallucination across multiple VLMs and benchmarks.

- The paper is well-written and technically sound.

**Weaknesses:**

1. The calculation of AttnRecall relies on the paired dataset, limiting its application on benchmarks without paired bounding boxes or segmentation masks.

2. The evaluation is limited on hallucination benchmarks. The ability of the proposed method in reducing hallucination in other real-world tasks (such as spatial reasoning).

**Questions:**

1. In the definition of A2,  why do you choose the question tokens with high standardized variability as the key question tokens?

2. Can your method improve the VLMs' performance in other visual-centric tasks such as spatial reasoning? It is suggested to evaluate your method in benchmarks including the RealWorldQA [1] and 3DSRBench [2]. If the performance is not improved, please provide some analysis about the underlying reason.

3. For VLMs with stronger spatial reasoning capability such as [3], will it show better AttnRecall?

[1] https://huggingface.co/datasets/visheratin/realworldqa

[2] Ma, Wufei, et al. "3dsrbench: A comprehensive 3d spatial reasoning benchmark." Proceedings of the IEEE/CVF International Conference on Computer Vision. 2025.

[3] AI, Inclusion, et al. "M2-reasoning: Empowering mllms with unified general and spatial reasoning." arXiv preprint arXiv:2507.08306 (2025).

---

> ### Author Response · Authors · 2025-11-20
> **[1/2] Author response**
>
> We sincerely thank you for your careful comments and thorough understanding of our paper! Here we give point-by-point responses to your comments and describe the revisions we made to address them.
> > **W1:** The calculation of AttnRecall relies on the paired dataset, limiting its application on benchmarks without paired bounding boxes or segmentation masks.
>
> Thank you for pointing out that the current calculation of AttnRecall depends on bounding box annotations. Your observation is correct, since AttnRecall measures how well the attended regions overlap with object regions.
>
> At the same time, AttnRecall is not restricted to datasets that already provide paired bounding boxes or segmentation masks. In practice, **many existing detectors and segmentors can produce reasonably accurate bounding boxes** for arbitrary image datasets. As a result, AttnRecall can be computed by combining our metric with such models.
>
> To demonstrate this, we conduct an additional experiment on the COCO validation set. We randomly sample 1000 images and compute AttnRecall for Ocean-r1 in two ways:
> (1) using the ground truth annotations, and
> (2) using bounding boxes predicted by a Mask R-CNN based segmentation model (40M parameters) [1].
>
> The results are summarized below.
>
> | seg model    | 0      | 6      | 12     | 18     | 24     | 27     |
> | ------------ | ------ | ------ | ------ | ------ | ------ | ------ |
> | ground truth | 31.05% | 32.64% | 48.56% | 53.79% | 50.92% | 49.23% |
> | mask r-cnn   | 32.05% | 33.61% | 49.38% | 52.72% | 48.89% | 47.61% |
>
> As shown, the trends based on Mask R-CNN predictions are very **close** to those based on ground truth, with only minor numerical differences. This empirical result suggests that AttnRecall can be reliably extended to datasets without paired annotations by leveraging existing detection or segmentation models.
>
> We have included the code for computing AttnRecall on arbitrary datasets using Mask R-CNN in the anonymous repository and updated the description of AttnRecall in the main text (lines 265–269), and will release the code publicly after acceptance.
>
> ---
>
> > **W2:** The evaluation is limited on hallucination benchmarks. The ability of the proposed method in reducing hallucination in other real-world tasks (such as spatial reasoning).
> > **Q2:** Can your method improve the VLMs' performance in other visual-centric tasks such as spatial reasoning? It is suggested to evaluate your method in benchmarks including the RealWorldQA and 3DSRBench. If the performance is not improved, please provide some analysis about the underlying reason.
>
> Thank you for the insightful suggestion. Following your recommendation, we further evaluated $A^4$-MLRM on RealWorldQA [2] and 3DSRBench [3]. We report accuracy on these benchmarks, comparing three reasoning models with and without $A^4$-MLRM:
>
> | data      | Ocean-r1 | + $A^4$-MLRM | Orsta  | + $A^4$-MLRM | MM-Eureka | + $A^4$-MLRM |
> | --------- | -------- | --------- | ------ | --------- | --------- | --------- |
> | RWQA      | 63.27%   | **70.33%**    | 65.23% | **67.84%**    | 60.39%    | **65.49%**    |
> | 3DSRBench | 37.66%   | **40.06%**    | 43.44% | **45.14%**    | 56.25%    | **59.24%**    |
>
> As shown, $A^4$-MLRM consistently improves performance on both RWQA and 3DSRBench across all three base models. These results demonstrate that our method not only reduces hallucination on dedicated hallucination benchmarks, but also enhances the models’ grounding in real-world visual scenes and spatial reasoning, and we have updated Appendix with these results (Table 7).
>
> ---
>
> > **Q1:** In the definition of A2, why do you choose the question tokens with high standardized variability as the key question tokens?
>
> We select question tokens with high standardized variability as key tokens because this directly follows Observation 2 (line 200–206). During multi-step reasoning, the model needs to collect clues, and those **more relevant** to the question are **repeatedly mentioned** and attended to during the \<reason\> and \<answer\> phases, which leads to larger fluctuations (higher variance) in their attention over time. As shown in Fig. 3 (bottom-left), the key tokens highlighted in the red box (e.g., “batter”, “helmet”) are exactly those repeatedly focused on during reasoning, and thus exhibit higher standardized variability than other question tokens.

---

> ### Author Response · Authors · 2025-11-20
> **[2/2] Author response**
>
> > **Q3:** For VLMs with stronger spatial reasoning capability such as M2-reasoning, will it show better AttnRecall?
>
> We thank the reviewer for your insightful suggestion. We follow your comment and compute AttnRecall on M2-Reasoning [4]. The results are:
>
> |                 | 0     | 6     | 12    | 18    | 24    | 27    |
> | --------------- | ----- | ----- | ----- | ----- | ----- | ----- |
> | M2-Reasoning-7b | 30.66 | 31.18 | 42.80 | 48.55 | 54.16 | 46.29 |
>
> We make two observations.
>
> 1. M2-Reasoning achieves relatively high AttnRecall. Among the six evaluated 7b models (R1-onevision, Ocean-r1, MM-Eureka, Orsta-r1, CoF, and M2-Reasoning), it **ranks second**, which is consistent with its stronger spatial reasoning ability.
> 2. The peak of AttnRecall for M2-Reasoning appears around layers 21 to 26, slightly **later** than other models whose peaks lie around layers 18 to 24. Recent work [5-6] suggests that later layers of large models focus more on reasoning and information organization. We hypothesize that the training of M2-Reasoning better integrates perceptual information into these later reasoning oriented layers, which leads to the shifted peak in AttnRecall.
>
> ## References
>
> [1] He, Kaiming, et al. "Mask r-cnn." ICCV 2017.
>
> [2] https://huggingface.co/datasets/visheratin/realworldqa.
>
> [3] Ma, Wufei, et al. "3dsrbench: A comprehensive 3d spatial reasoning benchmark." CVPR 2025.
>
> [4] AI, Inclusion, et al. "M2-reasoning: Empowering mllms with unified general and spatial reasoning." arXiv preprint arXiv:2507.08306 (2025).
>
> [5] Tao, Mingxu, et al. "Probing Multimodal Large Language Models for Global and Local Semantic Representations." LREC-COLING 2024.
>
> [6] Yu, Zhuoran, et al. "How Multimodal LLMs Solve Image Tasks: A Lens on Visual Grounding, Task Reasoning, and Answer Decoding." COLM 2025.

---

> > ### Comment · Reviewer_ndkQ · 2025-11-25
> > **Thanks for the Rebuttal**
> >
> > Thanks the authors for providing comprehensive rebuttal. My concerns regarding the methodology designation have been addressed. I will keep my positive score.

---

> > > ### Author Response · Authors · 2025-11-26
> > > **Thanks a lot for your time and effort!**
> > >
> > > We truly appreciate your careful effort throughout the review process and are very glad that we were able to address your concerns. Your feedback has been very helpful for refining the design of our methodology.

---

### Author Response · Authors · 2025-11-20
**General response to all reviewers**

We thank all reviewers for their constructive comments and for recognizing the contributions of our work, such as Reviewer ndkQ’s remarks on the “very interesting insights” and “well-written and technically sound” presentation, Reviewer XuFa’s appreciation of the “practical and deployment-friendly design” and “broader applicability,” and Reviewer htgT’s acknowledgement of the “significant advantage” and “substantial performance gains.”

In response to the reviews, we have **updated the PDF**: in particular, we **refactored the notation** by unifying inputs as $X$, outputs as $y$, and attention as $A$, added a **global notation summary** (Table 1), and **improved the exposition** of the Method section (Sec. 4). All changes are highlighted in **blue** in the updated manuscript.

We again thank all reviewers for their efforts, and we would be very happy to further clarify any remaining questions and continuously improve our paper.

---

### Author Response · Authors · 2025-11-30
**[1/2] Rebuttal Summary for ACs and SACs**

Dear **Area Chairs**, **Senior Area Chairs**, and **Reviewers**,

Thank you for your tremendous effort in handling the recent reviewer anonymity leak incident. We greatly appreciate the steps you have taken to uphold the integrity of the review process and your dedication to maintaining a fair research community.

To save your time and help you quickly understand the rebuttal process for submission 1335 (A$^4$-MLRM), we first **summarize the strengths** of our submission as recognized by the reviewers. We then **categorize the reviewers’ concerns into experimental issues and non-experimental issues**. A table is provided to summarize how we addressed all experimental issues, followed by detailed responses to each reviewer’s non-experimental concerns.

---

## Strengths Summary

All reviewers acknowledged that A$^4$-MLRM, as an **architecture-agnostic, training-free, and parameter-free plugin**, brings clear gains in reducing hallucination across models:
- **Reviewer ndkQ**: “A4-MLRM effectively suppresses the hallucination across multiple VLMs and benchmarks.”
- **Reviewer XuFa**: “Practical and deployment-friendly design: no training, no architectural modifications. Demonstrates consistent improvements across multiple MLLMs and evaluation benchmarks.”
- **Reviewer htgT**: “A4-MLRM is completely training-free and parameter-free, resulting in minimal deployment cost, a significant advantage over prior training-stage mitigation efforts; it’s also architecture-agnostic.”

Additionally, the reviewers praised our paper’s insights and broader impact:
- Reviewer ndkQ highlighted the “**very interesting insights**” of our work and found it “**well-written and technically sound**.”
- Reviewer XuFa noted that our study “**addresses a challenge**” in the field, with “**analysis and ablations supporting empirical choices**” and “**broader applicability**.”
- Reviewer htgT observed that A$^4$-MLRM “**achieves substantial performance gains**” and “**successfully transfers to non-reasoning MLLMs**.”

These comments underscore that A$^4$-MLRM’s design is widely applicable and effective in mitigating hallucinations without retraining or architecture changes.

---

## Experimental Issues

We group the reviewers’ experimental questions/weaknesses by topic and summarize how we addressed each in the rebuttal:

| Reviewer Issue and topic | How We Addressed It|
|-|-|
| Reviewer `XuFa W6 & Q5`;  `htgT Q1`: **Scalability to Larger Models** | We added experiments on three larger 32B models (MM-Eureka-32B, ORSTA-32B, and Qwen2.5-VL-32B). The results show that A$^4$-MLRM scales effectively to bigger model sizes, consistently improving performance. (Updated in Table 13 of the revision.)|
| Reviewer `ndkQ Q3`; `XuFa Q7`: **Generality Across Architectures** | We evaluated A$^4$-MLRM on diverse architectures and training paradigms, including M2-Reasoning (spatial reasoning-focused), CoF-RL (reinforcement learning zoom-in strategy), Reverse (self-verification), and also LLaVA-1.5 and LLaVA-Mistral. A$^4$-MLRM provided consistent gains, demonstrating strong generalization across model types. (Results added in Table 14 of the revision.)|
| Reviewer `ndkQ W2 & Q2`; `XuFa W3 & Q3`: **Broader Dataset Evaluation**| We extended our evaluation to more diverse scenarios: an open-ended subset of RH-Bench (long-form VQA), a real-world visual QA dataset (RealWorldQA), and a 3D spatial reasoning benchmark (3DSRBench). Across all these tasks, A$^4$-MLRM consistently improved model accuracy and reduced hallucination. (Results reported in Table 7 of the revision.)|
| Reviewer `XuFa W4 & Q2`: **Ablation & Sensitivity Analysis** | We conducted extensive ablations varying all hyperparameters. The performance remained stable across a broad range of $\tau_q$, $\tau_o$, $\tau_v$, and DBSCAN-$\epsilon$ values, indicating that our method is not overly sensitive to specific parameter choices. (Detailed results in Tables 8–11 of the revision.)|
| Reviewer `XuFa Q6`; `htgT Q2`: **Failure Case Examination**| We analyzed the cases where A$^4$-MLRM did not help. We found that truly A$^4$-MLRM-induced errors are extremely rare (only about 0.15% of the total evaluation set). We also provided representative failure case visualizations. (See Figures 11–12 in the revision.)|
| Reviewer `XuFa W5 & Q4`; `htgT W2`: **Efficiency and Cost Analysis** | We provided a detailed cost analysis for both offline and online modes, measuring latency and throughput under different Stage-1 output lengths. The results (in Fig. 5b and Appendix Table 12 of the revision) quantify the trade-off between output length and accuracy.|

---

> ### Author Response · Authors · 2025-11-30
> **[2/2] Rebuttal Summary for ACs and SACs**
>
> ## Non-Experimental Issues
>
> ---
>
> ### **Reviewer ndkQ**
>
> Reviewer ndkQ’s concerns centered on the methodology designation. In particular, they asked why we select question tokens with high attention variability as key tokens (Q1) and pointed out the dependency of AttnRecall on annotated data (W1). We addressed these as follows:
> - **Key question token (`Q1`)**: for Q1, we explained that choosing question tokens with high standardized variance follows directly from our Observation 2: such tokens correspond to concepts repeatedly focused on during reasoning, hence identifying key query terms.
> - **AttnRecall’s reliance on paired annotations (`W1`)**: we noted that our AttnRecall metric can be generalized using off-the-shelf detectors or segmenters (which we demonstrated by recomputing AttnRecall with Mask R-CNN predictions, showing negligible difference from using ground-truth boxes).
>
> Reviewer ndkQ considered our rebuttal comprehensive and felt that it resolved all of the reviewer’s concerns.
>
> ---
>
> ### **Reviewer XuFa**
>
> Reviewer XuFa raised important conceptual questions, mainly about our novelty vs prior work and the use of attention as a reliable signal:
> - **Comparison to prior approaches (`W1 & Q9`)**: We clarified how A$^4$-MLRM differs from prior attention-guided grounding/decoding strategies. Our approach is driven by a new metric (AttnRecall) and a **fine-grained token-level** analysis of attention across layers. We related this to recent work on the reasoning–perception trade-off, explaining that A$^4$-MLRM mitigates the degradation of visual grounding in long chain-of-thought scenarios by explicitly refocusing on visual evidence. Our new analysis shows that the method **increases the length of reasoning chains** and yields larger gains on reasoning-focused datasets than on general datasets.
> - **Reliability of attention as a signal (`W2 & Q1`)**: We acknowledged the concern that attention maps can be noisy or misaligned and provided justification for using attention in our method. We cited **qualitative and quantitative evidence** (Fig. 1b, Fig. 2b) and recent studies showing that VLMs often still attend to relevant regions even when they answer incorrectly. We further explained how A$^4$-MLRM denoises attention by averaging across heads within each layer (**reducing out-of-region variance by 8×**) and **applying noise-robust clustering** (DBSCAN) to isolate coherent high-attention regions, so that we retain meaningful signals while filtering out spurious ones.
>
> ---
>
> ### **Reviewer htgT**
>
> Reviewer htgT’s primary concerns were about the presentation and clarity of the paper (W1). We took the presentation feedback very seriously and made significant revisions:
> - **Presentation improvements (`W1`)**: We thoroughly revised the paper’s structure and writing for clarity. Specifically:
>     1. we **unified the notation** throughout the paper and added a global notation summary table in the Preliminaries;
>     2. we **moved our key empirical observations into a standalone Section 3 (Motivation)** to set up our method more clearly;
>     3. we **refactored the Method section** to walk the reader through the A$^4$-MLRM procedure step by step; and
>     4. we **simplified and redrew the figures** (especially the overview diagram in Fig. 3) to focus on the core workflow rather than overwhelming details.
>
> These changes are all aimed at making the paper significantly easier to read and follow.
>
> ---
> ## Summary
>
> Submission 1335 initially received an average score of **5.33**. Throughout the rebuttal, we have addressed every reviewer concern point-by-point and incorporated the necessary clarifications and results into the revised manuscript. For instance, Reviewer ndkQ explicitly noted that their concerns were fully addressed by our comprehensive answers.
>
> We are grateful that the reviewers’ thoughtful feedback helped us improve the paper’s clarity and reinforce its contributions. In particular, their input led us to demonstrate even broader applicability of A$^4$-MLRM and to highlight additional insights, such as its **effectiveness even on small objects** (`XuFa Q8`), its ability to **reduce the model’s perceptual burden and thereby free more capacity for reasoning** (`XuFa Q9`), and the **stability of our AttnRecall-based perception-layer findings** across different models (`ndkQ Q3`, `XuFa W6 & Q5` and `htgT Q1`).
>
> In conclusion, A$^4$-MLRM is an **architecture-agnostic, training-free, and parameter-free** plugin that can be applied to a wide range of large models to robustly reduce hallucinations. Together with the **insights** we provide into where and how perception and reasoning emerges in modern MLRMs, and the **point-by-point rebuttal** that addresses all reviewer concerns with new experiments and analyses, we hope that these aspects can be taken into account when forming an overall assessment of our submission.
>
> Thank you again for your consideration.
>
> Sincerely,
>
> Submission 1335 Authors

---

### Meta-Review · Area_Chair_wfiq · 2025-12-24

**Summary:**

The paper proposes a two-stage pipeline to quantify and mitigate hallucination in multimodal reasoning models. The major concerns on this paper include limited novelty compared with existing attention-guided hallucination mitigation methods (XuFa) and high computational cost of the method (XuFa, htgt). After carefully reading the paper, I agree with Reviewer XuFa and htgt regarding these concerns.

Specifically, the main insight of this paper seems to be very similar to published papers on the attention mechanism of MLLM hallucination, such as https://arxiv.org/pdf/2503.01773. Although the authors explained that this paper focuses on fine-grained token-level analysis in the reasoning setting, this distinction appears to be largely incremental. The two-stage pipeline introduces non-trivial computational overhead, and the paper lacks a clear cost–benefit analysis to justify the added latency compared with alternatives.

**Reviewer Concerns:**

The following concerns are addressed by the rebuttal, but the concerns regarding novelty, reliability and latency mentioned above are not fully addressed.

1. AttnRecall relies on paired datasets (ndkQ).

2. Evaluation is limited on hallucination benchmarks (ndkQ, XuFa).

3. Sensitivity to hyperparameters is not systematically studied (XuFa).

4. Experiment on larger models is lacking (XuFa).

5. Writing is not clear (htgT)

**Reviewer Scores:**

Reviewer ndkQ will keep the positive score as indicated by the response.

Reviewer XuFa may not increase the score as the major concerns on novelty and computational cost are not addressed.

Reviewer htgT's concern on the writing issues is addressed, but I don't think the computational cost is addressed. Optimistically, Reviewer htgT will increase to 6.

---

### Decision · Program_Chairs · 2026-01-26

Reject